# Pleiotropic Roles of Calmodulin in the Regulation of KRas and Rac1 GTPases: Functional Diversity in Health and Disease

**DOI:** 10.3390/ijms21103680

**Published:** 2020-05-23

**Authors:** Francesc Tebar, Albert Chavero, Neus Agell, Albert Lu, Carles Rentero, Carlos Enrich, Thomas Grewal

**Affiliations:** 1Departament de Biomedicina, Unitat de Biologia Cel·lular, Facultat de Medicina i Ciències de la Salut, Centre de Recerca Biomèdica CELLEX, Institut d’Investigacions Biomèdiques August Pi i Sunyer (IDIBAPS), Universitat de Barcelona, 08036 Barcelona, Spain; albertchavero@gmail.com (A.C.); neusagell@ub.edu (N.A.); carles.rentero@ub.edu (C.R.); enrich@ub.edu (C.E.); 2Department of Biochemistry, Stanford University School of Medicine, Stanford, CA 94305, USA; alulopez@stanford.edu; 3School of Pharmacy, Faculty of Medicine and Health, University of Sydney, Sydney, NSW 2006, Australia

**Keywords:** Calmodulin, KRas, Rac1, calmodulin-binding proteins, signalling

## Abstract

Calmodulin is a ubiquitous signalling protein that controls many biological processes due to its capacity to interact and/or regulate a large number of cellular proteins and pathways, mostly in a Ca^2+^-dependent manner. This complex interactome of calmodulin can have pleiotropic molecular consequences, which over the years has made it often difficult to clearly define the contribution of calmodulin in the signal output of specific pathways and overall biological response. Most relevant for this review, the ability of calmodulin to influence the spatiotemporal signalling of several small GTPases, in particular KRas and Rac1, can modulate fundamental biological outcomes such as proliferation and migration. First, direct interaction of calmodulin with these GTPases can alter their subcellular localization and activation state, induce post-translational modifications as well as their ability to interact with effectors. Second, through interaction with a set of calmodulin binding proteins (CaMBPs), calmodulin can control the capacity of several guanine nucleotide exchange factors (GEFs) to promote the switch of inactive KRas and Rac1 to an active conformation. Moreover, Rac1 is also an effector of KRas and both proteins are interconnected as highlighted by the requirement for Rac1 activation in KRas-driven tumourigenesis. In this review, we attempt to summarize the multiple layers how calmodulin can regulate KRas and Rac1 GTPases in a variety of cellular events, with biological consequences and potential for therapeutic opportunities in disease settings, such as cancer.

## 1. Introduction

Calmodulin is a ubiquitously expressed small protein (148 amino acids) and considered the most important Ca^2+^ sensor in non-muscular cells [1,2]. This Ca^2+^-sensing function allows calmodulin to act as a signalling molecule, translating transient fluctuations of Ca^2+^ levels inside cells into rapid and appropriate alterations of a vast number of cellular activities. Calmodulin-mediated cellular responses commonly occur through its interaction with a plethora of effectors named calmodulin-binding proteins (CaMBPs), most of them in a Ca^2+^-dependent manner [1,3,4]. Multiple calmodulin interaction motifs within CaMBPs exist, comprising basic amphiphilic helices, a cluster of polybasic amino acids, attached prenyl groups, all of which are found in small GTPases (see below), as well as the widely distributed ilimaquinone (IQ) motif [3,5,6,7,8]. Up to date, more than 450 calmodulin interacting proteins in human and mice have been described (BioGRID: https//thebiogrid.og; [9]), suggesting multiple cellular pathways that are triggered or regulated by calmodulin. This large and diverse group of CaMBPs includes enzymes such as kinases and phosphatases, ion channels, and cell surface receptors. Most relevant for this review, calmodulin also binds to members of the Ras superfamily of small GTPases that control cellular processes like endo- and exocytosis, protein and vesicle transport, cytoskeleton organization and dynamics, proliferation, and migration [10,11]. Ras GTPases are divided into five large subfamilies: Ras, Rho, Ran, Arf, and Rab [12]. A substantial number of these small GTPases are calmodulin-binding proteins, pointing at their central role to communicate changes in the local microenvironment and facilitate a rapid cellular response upon transient Ca^2+^ elevation.

In the following section, we will give a brief overview on calmodulin-binding GTPases (Table 1). Calmodulin acts on these GTPases either via direct binding or indirectly through interaction with CaMBPs that represent regulators of GTPases. Consequently, these interactions control the spatiotemporal localization, activity, and biological outcome of GTPase signalling. Small GTPases are molecular switches that are inactive when bound to GDP and active when bound to GTP. This cycle of GTPase activation and inactivation is controlled by guanine nucleotide exchange factors (GEFs) and GTPase activating proteins (GAPs), respectively. GEF proteins stimulate GDP dissociation, enabling the GTPase to incorporate GTP, which is 10-fold more concentrated than GDP in the cytosol. This turns the GTPase into an active state, capable of interacting with effectors to activate pathways that determine multiple aspects of cellular behavior. GTPase signalling is then terminated by GAP proteins, which enhance GTP hydrolysis through the activation of their intrinsic GTPase activity [13,14,15]. Calmodulin can positively or negatively modulate a variety of GEFs and GAPs, leading to GTPase activation or inhibition depending on the subset of GTPases analyzed. These multiple interactions of calmodulin with GTPases, GEFs and GAPs have made it difficult to dissect and then define calmodulin functions in regard to specific signalling pathways or biological outcomes. However, the use of more sophisticated in vitro protein–protein interaction techniques, pharmacological inhibitors, including small molecules or inhibitory peptides, and state-of-the-art imaging techniques, now provides a clearer picture of how calmodulin regulates different proteins of the Ras GTPase superfamily.

The present review endeavors to compile the current understanding, obtained over more than two decades, summarizing the role of calmodulin in the regulation of small GTPases. We will specifically focus on K-Ras4B (thereafter KRas) and Rac1, which both bind calmodulin in a Ca^2+^-dependent manner [16,17,18,19,20,21,22,23]. Adding complexity, calmodulin regulates signalling outputs from these GTPases indirectly by binding and influencing the localization and activity of other CaMBPs that act as GTPase regulators or effectors. Calmodulin-dependent GTPase signalling is coupled to pathological settings, as KRas and Rac1 are often drivers of signalling pathways that critically stimulate tumour formation and progression [24,25,26]. The latter has been recently reviewed in more detail in a previous issue of this journal [27,28].

## 2. Interaction of Calmodulin with GTPases

Up to date, several GTPases have been identified as Ca^2+^-dependent CaMBPs, including KRas, RalA, RalB, Rin (RIC in *Drosophila*), Kir/Gem, Rab3A, Rab3B, Cdc42, and Rac1 [16,22,23,29,30,31,32,33,34,35,36] (Table 1). Most of the abovementioned GTPases interact with Ca^2+^/calmodulin through a C-terminal polybasic region (PBR) and a post-translationally incorporated prenyl group within this region that provides the lipid anchor to direct those GTPases to cell membranes.

### 2.1. RalA and RalB

RalA and RalB isoforms are both Ras effectors [37] that have a second motif located in the N-terminal domain that interacts with calmodulin [34,38,39]. The interaction of calmodulin with these two Ral GTPases appears to be independent of the Ral-GTP or -GDP status, yet necessary for Ral activation by thrombin in human platelets [38]. Although the biological outcome and significance of these interactions is not completely understood, calmodulin binding to RalA increases its GTP loading and blocks its phosphorylation by protein kinase A (PKA), protein kinase G (PKG) and protein kinase C (PKC) [40,41]. Hence, a role for calmodulin as a GEF for RalA and Rac1 GTPases has been hypothesized [21,40]. Along these lines, calmodulin can also inhibit PKC-mediated serine 181 (Ser181) phosphorylation of KRas, which allows the modulation of KRas function and will be discussed in more detail below (Section 3) [16].

### 2.2. Rab3A and Rab3D

Rab3A is found on secretory vesicles of neuronal and endocrine cells that regulate neurotransmitter and hormone release [42,43,44,45]. Overexpression of a constitutively active Rab3A mutant (Q81L) inhibits Ca^2+^-induced exocytosis in pancreatic β-cells. In these cells, calmodulin preferentially binds GTP-bound Rab3A in a Ca^2+^-dependent manner, and calmodulin dissociation from Rab3A can be triggered by glucose-induced insulin secretion [42]. Accordingly, calmodulin/Rab3A interaction may ensure proper physiological response and prevent secretory events in pancreatic β-cells linked to insulin release in the absence of elevated plasma glucose levels [30]. Upon enhanced glucose levels and the subsequent rise in Ca^2+^ levels in pancreatic β-cells, Ca^2+^/calmodulin may dissociate from Rab3A. This could be due to competing CaMBPs with a higher calmodulin affinity, for example Ca^2+^/calmodulin activated protein kinase II (CaMKII). The preference for calmodulin to bind CaMKII during granule secretion could be the underlying cause for increased insulin release [42]. Additionally, the Rab3D isoform involved in exocytosis in mast cells, osteoclasts and pancreatic β-cells has recently been identified as a Ca^2+^-dependent CaMBP. Inhibiting calmodulin/Rab3D interaction with calmidazolium chloride diminished osteoclastic bone resorption [46].

### 2.3. Rac1 and Cdc42

Rac1 and Cdc42 also represent GTPases of the Ras superfamily that bind calmodulin [22,23] (Table 1). Similar to the calmodulin/KRas interaction modules [16], several domains facilitate these protein interactions. This includes a region located between amino acids 151–164 that can adopt an alpha helix conformation, as well as the polybasic domain together with the adjacent prenyl group at the C-terminus. Rac1/calmodulin interaction and its regulatory circuits and downstream effectors will be explained in more detail in Section 4. Inhibition of calmodulin has opposite consequences for Rac1 and Cdc42 activity, leading to decreased Rac1-GTP, but elevated Cdc42-GTP levels [23]. Calmodulin can control Cdc42 activity and Cdc42-mediated regulation of the actin cytoskeleton through binding of the IQ Motif Containing GTPase Activating Protein 1 (IQGAP1), a well-known Cdc42 regulator [47]. IQGAP1 binds Cdc42 and keeps the GTPase bound to GTP (active), which concomitantly enhances F-actin crosslinking activity of IQGAP1 [48,49]. Ca^2+^/calmodulin, through interaction with IQGAP1, dissociates IQGAP1 from Cdc42 and negatively regulates this GTPase (see section 3.2.2 for further details) [50,51].

IQGAP also binds the ubiquitous GTPase Rap1, which contributes to cell adhesion, synaptic plasticity, and mitogen-activated protein kinase (MAPK) pathway activation. Although Rap1 has yet to be identified as a CaMBP, IQGAP1 binding to Rap1, in contrast to other GTPases interacting with IQGAP1, attenuates its activation triggered by cAMP, fibronectin, or collagen I. Therefore, calmodulin may trigger dissociation of IQGAP1 from Rap1, possibly due to calmodulin and Rap1 competing for IQGAP1 binding, leading to increased Rap1 activity [52]. Accordingly, pharmacological inhibition of calmodulin blocks Rap1 (and Ras) activation after depolarization in cortical neurons. Consequently, activation of the MAPK pathway downstream of Ras and Rap1 is inhibited, making calmodulin a crucial and common regulator of signalling events driven by these two GTPases in neurons [53].

### 2.4. Ric and Rin

There are members of the Ras superfamily that lack the prenylation motif (also termed CAAX box), which commonly enables membrane association of most GTPases, as is the case for the RGK and Rit subfamilies [32,35,54] (Table 1). These two GTPase subfamilies rely on long stretches of basic amino acids for plasma membrane localization by electrostatic interaction with acidic membrane phospholipids. Rin is a GTPase of the Rit subfamily that is expressed in neurons and necessary for neurite growth [55]. This protein has a polybasic cluster and several hydrophobic residues within a C-terminal 25 amino acid region, allowing interaction with calmodulin in a Ca^2+^-dependent manner [32,54]. Although the molecular consequences of this interaction are still unclear, blocking calmodulin inhibits the ability of Rin to promote neurite growth [55] and attenuates its activation by growth factor receptors [54]. RIC, the ortholog of Rin in *Drosophila*, also interacts with calmodulin through a sequence located at position 242–261 [35]. The expression of an active RIC mutant altered the development of adult wing vein structures. This phenotype was intensified when calmodulin levels were reduced. Thus, interaction of calmodulin with RIC seems to negatively regulate this GTPase [56].

### 2.5. Rad, Kir/Gem, and Rem

Within the RGK GTPase subfamily, Rad, Kir/Gem, and Rem are engaged in the cell surface delivery and inhibition of Ca^2+^-channels, as well as the control of cell morphology and actin cytoskeleton remodeling by interfering with the Rho/Rho-kinase pathway [57,58,59] (Table 1). Calmodulin interacts with these GTPases in a Ca^2+^-dependent manner through their C-terminal domains. These C-termini are extended by 31 amino acids compared to Ras, containing polybasic and hydrophobic residues that are typical hallmarks for calmodulin binding [31,60]. Complex formation of calmodulin with Kir and Gem GTPases significantly inhibits their GTP binding [31]. Likewise, calmodulin associates preferentially with Rad-GDP [60]. In this GTPase, the C-terminal region is required for plasma membrane location [58]. Calmodulin, in a cooperative and competitive fashion together with 14-3-3 proteins, regulates the subcellular location of Kir/Gem and Rem2 [61,62]. Mutant versions of these GTPases that are defective in calmodulin binding have been preferentially found inside the nucleus. This change of location may explain the inhibition of voltage-dependent Ca^2+^-channels and determine cell-shape remodelling [61,63,64]. Beguin et al. demonstrated that binding of Ca^2+^/calmodulin to Kir/Gem maintains their cytoplasmic localization, which is necessary for their inhibitory effect on surface transport of Ca^2+^ channels. In such cytoplasmic locations, activated Kir/Gem (GTP-bound), by interacting with beta-subunits of the channel, impairs the physical association with alpha1-subunits and its traffic to the plasma membrane to function as a Ca^2+^ channel [63]. In addition, calmodulin prevents importins to bind nuclear localization signals within the C-terminal region of these GTPases and interferes with their nuclear translocation [64]. Finally, putative serine-phosphorylated residues neighboring nuclear translocation signals are consensus sites for PKC, PKA and protein kinase B (Akt) kinases. This could affect importin and possibly calmodulin binding to Kir/Gem and cytoplasmic and nuclear shuttling. Interestingly, phosphorylation of the Rad member of the RGK subfamily by PKC or casein kinase II abolishes the interaction of Rad with calmodulin [65].

## 3. Ras-GTPases and their Post-Translational Modifications Fine-Tune Cellular Signalling

The family of Ras GTPases includes three members: Harvey Ras (HRas), Neuroblastoma Ras (NRas), and Kirsten Ras (KRas), which has two splice variants, KRas-4A and the more prominent KRas-4B isoform (named here KRas). Ras proteins have the vital task to decode a variety of extracellular stimuli. At the cell surface, these stimuli are commonly derived from growth factor-activated tyrosine kinase receptors, surface receptors that participate in cell-extracellular matrix communications or receptors that modulate Ca^2+^-channels. Ras GTPases are ubiquitously expressed and regulate fundamental cellular endeavors like proliferation, differentiation, survival, apoptosis and cell mobility [75,76,77,78,79] that must be tightly regulated in a timely manner. Hence, constitutive or hyperactive Ras activity due to specific point mutations or overexpression is often associated with cancer initiation and progression or developmental defects.

### 3.1. The Prominant Role of KRas in Human Cancers

Ras proteins are mutated in 30% of human cancers, with 98% of these mutations being missense mutations in one of three hotspot residues at position G12, G13, and Q61. Amino acid substitutions at these positions prevent efficient Ras-GTP hydrolysis, maintaining Ras proteins in a constitutively active GTP-bound state. Mutations within the Ras family are not evently distributed, with KRas mutations found in 85% of these cancers, followed by NRas (11%) and HRas (4%) [76,80,81,82,83,84].

This indicates prominant roles for KRas mutations in cancer, which are most frequent in pancreatic (<60%), colon (<30%), and lung (<15%) cancers. NRas mutations are prominant in skin (<25%) and hematological (<15%) cancers [82] and HRas mutations often exist in head and neck cancers. This distribution of oncogenic Ras mutations is not well understood and may reflect tissue-specific expression patterns or different potentials to activate shared downstream pathways such as the MAPK and phosphoinositide 3-kinase (PI3K) signalling cascades. Based on the very high (~90%) homology among all Ras isoforms, these findings were initially difficult to interpret. However, over the last two decades, the hypervariable region (HVR), which consists of 24–25 residues at the C-terminus of Ras proteins and shares less than 20% homology within Ras isoforms, is considered to be most relevant. Whereas the N-terminal region (G domain, residues 1–165) is almost identical in Ras isoforms and contains the GTP/GDP-binding site (switch I and II domains), the HVR region is responsible for the anchoring of Ras proteins at the plasma membrane. Biochemical studies and high resolution imaging revealed that the diversity of HVR regions enabled the targeting of Ras isoforms to different microdomains at the plasma membrane. This differential Ras partitioning and nanoclustering within the plasma membrane is now well believed to strongly influence the strength and duration of signals generated, with consequences for signal transmission and output [85,86,87,88,89,90]. Altogether these studies identify HVRs of each Ras isoform to contain unique functions in normal physiological processes as well as in pathogenesis [91,92,93,94].

### 3.2. The Membrane Association of Ras Proteins

Many research efforts have aimed to dissect the molecular events that ensure HVR-mediated membrane association of Ras proteins and the various steps in Ras membrane insertion are now quite well understood. After Ras synthesis in the cytosol as a hydrophilic protein, several posttranslational modifications at the HVR domain provide the acquisition of signals for membrane insertion and the control for Ras isoform localization. In the cytosol, Ras proteins are firstly farnesylated at a cysteine residue in the C-terminal CAAX motif (C, cysteine; A, aliphatic residue; X, any residue) by a cytosolic farnesyl transferase. This acquisition of a farnesyl group (15-carbon isoprenyl) allows Ras insertion into the endoplasmic reticulum (ER) membrane, where AAX is subsequently subjected to proteolysis and the remaining cysteine is methylated by Ras-converting enzyme 1 (Rce1) and Isoprenylcysteine carboxyl methyltransferase (Icmt), respectively [95]. HRas, NRas, but also the less well-studied KRas-4A isoform, are further modified by palmitoylation, which allows their transport through the Golgi apparatus and subsequent exocytocic vesicular transport routes to isoform-specific plasma membrane localizations. In contrast, KRas is not palmitoylated, but requires a different second signal for membrane localization adjacent to the farnesylated cysteine, that consists of a cluster of polybasic amino acids, the PBR domain. This PBR region exerts a strong electrostatic interaction with acidic phospholipids of the cytosolic membrane leaflet [96,97,98,99]. The molecular mechanism responsible for KRas transport from the ER to the plasma membrane are still not fully understood, and the function of a chaperone protein proposed to hide the farnesyl group after synthesis in the ER have yet to be clarified [93,94,96].

The variety of posttranslational modifications seem to provide the basis for the different microdomain localizations of Ras isoforms. KRas differs substantially from H- and NRas in regard to their trafficking through cellular compartments, in particular in their dynamic association with membranes of the endocytic compartment [100]. During membrane turnover and/or growth factor-induced internalization, all Ras isoforms at the plasma membrane are transported through early endosomes. However, only KRas trafficks through the entire endocytic pathway, including late endosomes/multivesicular bodies, on its path to lysosomes for degradation [101]. Internalized Ras proteins can remain active or get activated and then signal from endosomes, creating additional signal diversity. This aspect has been reviewed in detail previously by our group and others [93,94,95,102,103,104,105,106].

KRas is the only Ras isoform that binds calmodulin [16,17,18,19,20,68,107,108,109,110], confering KRas-specific functions and biological outcomes. The association of calmodulin with KRas and its functionality will be discussed in more detail in the next two sections.

### 3.3. Domains Responsible for the Interaction of Calmodulin with KRas

Pioneering work by Villalonga and coworkers demonstrated, using affinity chromatography, that calmodulin was able to bind KRas, but not H- or NRas isoforms, from lysates of NIH3T3 fibroblasts. This direct protein–protein interaction occurred in a Ca^2+^-dependent manner and only when KRas was GTP-bound (active) [19]. In this initial study, it was proposed that binding of calmodulin to KRas in vivo inhibited downstream activation of the Raf and MAPK signalling cascade. Given the high incidence of KRas mutations in human cancers (see above), several groups aimed to dissect the molecular elements required for KRas/calmodulin interaction. Utilizing a wide range of methodologies and techniques such as affinity chromatography, immunoprecipitation, confocal microscopy, fluorescence spectroscopy and isothermal titration calorimetry (ITC), yeast two-hybrid systems, nuclear magnetic resonance (NMR), surface plasmon resonance spectroscopy, artificial liposomes and nanodiscs biomembranes, as well as computational simulations, the KRas/calmodulin interaction was thoroughly established over the years in numerous in vitro and in vivo models and cell types [16,17,18,19,20,68,107,108,109,110,111].

Some discrepancies and variations regarding the requirement of the GTP- or GDP-bound form of KRas, as well as the KRas domains mandatory for the interaction with calmodulin, have been reported. However, it is now well believed that the C-terminal HVR domain of KRas is essential to bind Ca^2+^/calmodulin. Indeed, the distinctive features of the KRas HVR region, lacking a posttranslational C-terminal palmitoylation found in the other Ras isoforms, but having the polybasic PBR domain, makes KRas the sole Ras isoform capable to bind calmodulin. Vice versa, in H- and NRas, the absence of basic amino acids in the HVR domains and the presence of the palmitoyl group sterically disturbs an interaction with calmodulin. Depalmitoylated KRas-4A containing a polybasic HVR region smaller than KRas may also bind Ca^2+^/calmodulin [112].

Several studies demonstrated that both the PBR domain and the farnesyl group within the HVR domain of KRas are essential to bind Ca^2+^/calmodulin [111,113,114,115,116]. López-Alcalá and coworkers further showed that the α-helix between residues 151–166 and the Switch II region, both located upstream the C-terminal HVR domain, also participates in the association with Ca^2+^/calmodulin [16]. A computational model, with support from experimental data, also supported the catalytic domain of KRas interacting with the N-terminal domain of calmodulin and the PBR wrapping around the calmodulin linker domain. In this model, the KRas farnesyl group docked into the C-terminal pocket of calmodulin [108,110]. NMR and ITC analysis confirmed the PBR domain of KRas as the primary binding site for the calmodulin C-terminal and linker domains. In addition, the catalytic GTP-bound domain of KRas aids the interaction via binding to the N-terminal domain of Ca^2+^/calmodulin [17]. In this study, it was hypothesized that the farnesyl group of the HVR domain may further enhance the binding affinity of KRas for Ca^2+^/calmodulin, which indeed was later demonstrated by others [18,117]. Analyzing different fragments and mutants of KRas bound to nanodisc biomembranes mapped the last six C-terminal residues of KRas together with the farnesyl group (KSKTKC-prenyl) as the minimal Ca^2+^/calmodulin-binding motif [68]. The identification of KRas forming a 2:1 stoichiometric complex with calmodulin suggested that KRas could first interact with the C-terminal domain of calmodulin. This would lead to a conformational change that would then allow the binding of a second KRas protein to the N-terminal domain of calmodulin [68]. Together with findings from several other studies, this suggested that calmodulin could participate as a molecular carrier to transport KRas, independent of its GTP- or GDP-bound form, from the ER to the plasma membrane. Alternatively, this function of calmodulin could translocate KRas from the plasma membrane to intracellular membrane compartments in a Ca^2+^-dependent manner. It has yet to be determined to what extend an abilty of calmodulin to sequester the polybasic-prenyl motifs of KRas could contribute to the overall membrane-associated trafficking of the entire pool of KRas proteins in each cell. Besides vesicular KRas trafficking, cytosolic KRas transport mechanisms also exist [100,118,119]. This transport route requires shielding the hydrophobic farnesyl group attached to the KRas C-terminus from the hydrophilic cytosol. This could be facilitated by calmodulin, but also others, such as prenylated rab acceptor 1 and phosphodiesterase 6 δ [67,118,119,120]. Together with fluctuations of cellular GTP/GDP ratios, and the reduced affinity of calmodulin towards inactive KRas (KRas-GDP), it remains to be clarified how these different transport routes partition in the cellular movement of KRas proteins [20,67,68,83,109] (Figure 1).

#### 3.3.1. Outcomes of KRas/calmodulin Interactions in Cell-Based Studies

In support of Ca^2+^/calmodulin triggering changes in KRas localization, Ca^2+^/calmodulin binds to KRas in platelets and MCF-7 cells and dissociates KRas from membranes [20]. In hippocampal neuronal cultures, glutamate activation of N-methyl-D-aspartate (NMDA) receptor induces an elevation of Ca^2+^ levels. This is associated with the translocation of active (KRas-GTP) and inactive KRas (KRas-GDP) from the plasma membrane to the Golgi apparatus and early endosomal membranes [67]. In both cases, KRas translocation is inhibited by the calmodulin inhibitor W7, strongly implicating Ca^2+^/calmodulin in this process [20,67]. In HeLa cells, W7 also reduces rapamycin-induced dissociation of the constitutively active KRas G12V mutant from the plasma membrane. In contrast, this W7-sensitive dissociation does not occur with the inactive KRas S17N mutant [118], which is consistent with the concept that calmodulin selectively recognizes and interacts with the activated form of KRas [19].

The binding of calmodulin to KRas may not always contribute to KRas translocation from the plasma membrane to endocytic compartments or participate in KRas transport from the ER to the plasma membrane [16]. In these studies, KRas mutants defective in calmodulin interaction, but with proper plasma membrane location as well as GTP-binding capacity, translocated to endosomal membranes or the Golgi apparatus after Ca^2+^ elevation in striatal neurons. Hence, in this setting, Ca^2+^-dependent KRas transport occured independent of calmodulin binding. This raised the possibility that the loss of KRas translocation observed by others [20,67], using the calmodulin inhibitor W7, may be related to an indirect effect of calmodulin on KRas trafficking [16]. Ca^2+^-insensitive factors, scaffold and carrier proteins influencing KRas localization also need to be considered. For example, in NIH3T3 cells, the binding of calmodulin and KRas in the presence of a Ca^2+^-ionophore was prominent at the plasma membrane, with only minor amounts of KRas/calmodulin complexes present in intracellular compartments. Along these lines, a KRas mutant defective in calmodulin binding was found predominantly at the plasma membrane [16]. This suggests that calmodulin may not always be essential for KRas transport from the ER to the plasma membrane, implicating other proteins to facilitate KRas transport independent of KRas/calmodulin complex formation. These observations raised the hypothesis that calmodulin could be a key element for selected KRas protein pools to generate discrete microdomains at the plasma membrane. This was indeed supported by Barceló and coworkers [121], providing an exciting model that would allow calmodulin to selectively regulate KRas signalling output.

#### 3.3.2. Additional Insights on the Nucleotide Dependence of KRas/calmodulin Interaction

A notable issue that remains controversial within the field is the nucleotide dependence of the KRas/calmodulin interaction. The majority of earlier reports support Ca^2+^/calmodulin specifically and preferentially binding active KRas (KRas-GTP) [16,17,18,19,29,117]. The globular domain of KRas bound to GDP resembles an autoinhibited conformation, as this sequesters the HVR region [122], thereby preventing interaction of KRas with calmodulin. Yet, several more recent studies favor nucleotide-independent interaction of these two proteins, irrespective of the GDP- or GTP-bound state of KRas [20,67,68,83,109]. To explain these different experimental outcomes remains difficult, but one plausible scenario could be the loss of the autoinhibited KRas-GDP conformation. Conformational changes may lead to differential sensitivity towards the GTP/GDP ratio in the different models and experimental settings analyzed, possibly allowing the HVR domain to interact with Ca^2+^/calmodulin, even when GDP is bound to KRas. Nevertheless, further experiments are required to validate the preferential interaction of calmodulin with active KRas in live cells.

Despite these still unresolved matters and based on the currently available literature, the HVR domain of KRas is critical for the interaction with Ca^2+^/calmodulin. This interaction can prevent PKC-mediated Ser181 phosphorylation in the PBR domain of KRas. Vice versa, phosphorylation of KRas by PKC at this Ser181 residue blocks Ca^2+^/calmodulin to interact with KRas [16,29,107]. This interplay between PKC-mediated KRas phosphorylation and KRas/calmodulin interaction is an imperative mechanism to modulate KRas activity and downstream signalling [123] and will be discussed in the next section.

### 3.4. Calmodulin-Dependent KRas Activity—Diverse Outcomes for Cellular Signalling in Normal and Oncogenic Settings

Ras activation at the plasma membrane requires the recruitment of GEFs to promote guanine nucleotide exchange and subsequent association with effectors, in particular Raf1 and PI3K signalling cascades. The serine/threonine kinase Raf1 is vital for MAPK activation that is in command of cell proliferation and differentiation [75,124,125,126,127], while PI3K is upstream of Akt signalling cascades and the Rac1 GTPase, both dictating cell survival, proliferation, and mobility [128,129,130].

To examine this further, the small-molecule calmodulin inhibitors N-(4-aminobutyl)-5-chloro-2-naphthalenesulfonamide, also termed W13, its analog W7, and other related compounds have been utilized widely. This identified Ca^2+^/calmodulin to modulate Ras activity and downstream effectors by a complex interconnected network of signalling routes. Overall, Ca^2+^/calmodulin can act at multiple levels to influence Ras activity. This includes regulating upstream receptor tyrosine kinase activity at the cell surface, to controlling recruitment and activity of various GEF and GAP proteins or as discussed in Section 3.1, by a direct interaction with the KRas isoform (Figure 2).

#### 3.4.1. Calmodulin Regulates Several Guanine Nucleotide Exchange Factors (GEFs) and GTPase Activating Proteins (GAPs) in Neuronal Cells

With regard to the impact of calmodulin on GEF and GAP proteins, a lot of current knowledge is based on studies from cells derived from the central nervous system. Adult neurons express substantial amounts of RasGRF, a dual GEF for Ras and Rac1. The Ras GTPase-activating protein SynGAP is highly enriched in postsynaptic density fractions. Both proteins are involved in Ca^2+^/calmodulin-dependent activation of the Ras/MAPK pathway [131,132,133,134,135,136]. RasGRFs (RasGRF1 and RasGRF2) are activated by Ca^2+^/calmodulin binding to their N-terminal IQ motif and together with RasGRF phosphorylation, glutamate-induced Ca^2+^ increase can trigger MAPK activation [132,134,135,136,137]. In contrast, CaMKII-mediated phosphorylation of SynGAP increases its GAP activity thereby inhibits Ras and downstream effectors [133,138,139]. Calmodulin can activate both RasGRF1 and RasGRF2 in the hippocampus, although both GRFs mediate opposite forms of synaptic plasticity. RasGRF2 is believed to mediate the ability of subpopulations of the heterodimeric NMDA receptors to promote long-term potentiation. On the other hand, CaMKII-mediated phosphorylation and activation of SynGAP is associated with other pools of NMDA receptors that allow switching the preference of RasGRF1 from Ras to Rac1 signalling, which thereby promotes long-term depression [137].

#### 3.4.2. Calmodulin Regulates Mitogen-Activated Protein Kinase (MAPK) and Phosphoinositide-3 Kinase (PI3K) Signalling in Non-neuronal Cells

In non-neuronal cells, the modes of calmodulin-dependent Ras inactivation are very diverse and often differ depending on the cell line analyzed, with COS1, CHO, NIH3T3, Swiss3T3, HeLa, and NRK cells being the best-studied cell types to date [19,107,140,141,142,143,144,145,146].

At the outer leaflet of the plasma membrane and upstream of Ras GTPases, Ca^2+^/calmodulin can inhibit shedding of membrane-anchored growth factors via matrix metalloproteases [147,148]. Consequently, the calmodulin inhibitors W13 and W7 allow increased release of soluble growth factors that potently activate tyrosine kinase receptors like the epidermal growth factor receptor (EGFR) in an autocrine-paracrine manner [140,141,146,149]. EGFR activation by calmodulin inhibitors can be further enhanced as interaction of Ca^2+^/calmodulin with EGFR inhibits its tyrosine kinase activity [150,151,152]. In addition, CaMKII-mediated EGFR phosphorylation inhibits EGFR tyrosine kinase activity [153]. Blocking these multiple modes of calmodulin-mediated EGFR downregulation effectively permits Ras activation, followed by the recruitment and activation of Raf-1 and the MAPK pathway. This is potentiated further, as Ca^2+^/calmodulin can bind and not only prevent PKC-mediated KRas Ser181 phosphorylation, but also cause KRas release from the plasma membrane to the cytosol. Both events downregulate KRas-dependent activation of downstream effectors, such as the MAPK pathway (see Section 3.1) [20,67,68,83,107,121,123,141,154]. Hence, calmodulin inhibition leads to KRas activation, followed by Raf1/MAPK activation. Blocking calmodulin can promote Raf/MAPK signalling, although in serum-starved NIH3T3, Swiss3T3, NRK, A431, and HeLa cells, this occured most likely irrespective of tyrosine kinase receptors stimulation [19,144]. Specific expression pattern or preferred signalling routes seem to exist, as inhibition of calmodulin in COS1, CHO, NR6, HEK293, hepatocytes, and PC12 exerted an inhibitory effect on MAPK signalling [140,142,143,155,156]. The different outcomes of calmodulin inhibition on MAPK signalling have been summarized in Table 2.

This differential reliance on dissimilar signalling routes is reflected in studies that compared calmodulin-mediated PI3K signalling in COS1 and NIH-wt8 cells. Despite calmodulin being considered necessary for PI3K activation [142,143,159,160,161], this is only critical in COS1 cells to ensure Raf1 and MAPK activation [142]. These diverse outcomes may reflect the differential involvement and contribution of other Ras isoforms besides KRas in overall MAPK signal output in the various cell lines. In fact, H- and KRas expression levels, GTP loading, and Raf1 interaction in COS1 and NIH3T3-wt8 cells suggests that the overall role of calmodulin in MAPK signal output is determined by the ratio of activated H- and KRas and the cell-specific contribution of each isoform in Raf-1 activation [142].

The magnitude of interaction between KRas-GTP and calmodulin may be a key element in the modulation of Ras/Raf1/MAPK signalling in a PI3K-independent manner. In fibroblasts, calmodulin prevents activation of KRas and inhibition of GAPs targeting KRas by PKC [29,107,123], which is crucial to elicit Raf1/MAPK activation when calmodulin was inhibited [29,107]. Therefore, balancing interaction of KRas with either PKC or calmodulin is pivotal to regulate Ras effectors that control proliferation, survival and migration [123]. Based on initial studies proposing PKC-mediated Ser181 KRas phosphorylation [162], this was then confirmed in vitro and in vivo to establish Ca^2+^/calmodulin directly binding to KRas to modulate its signalling capacity [16,29,107].

#### 3.4.3. Calmodulin and Ser181 KRas Phosphorylation

PKC or calmodulin interacting with KRas have consequences for the cellular location of KRas, as Ser181 phosphorylation alleviates the electrostatic interaction with the plasma membrane. This causes partial relocation of active KRas to endocytic compartments, inducing specific KRas signalling outcomes from intracellular sites [119,154,163,164]. Work from our laboratories revealed that KRas can elicit Raf1/MAPK activation from the endocytic compartment. Although these signalling events seem to occur irrespective of KRas phosphorylation, it leads to prolonged MAPK signalling that initiates at the plasma membrane [101].

Given the profound effects of Ser181 phosphorylation on KRas signalling, phosphomimetic, and non-phosphorylatable oncogenic KRas mutants (KRasG12V-S181D, KRasG12V-S181A) were studied extensively. The phosphomimetic KRasG12V-S181D mutant displayed increased MAPK and Akt activity at low growth factor concentration that correlated with increased foci formation, cell mobility and resistance to apoptosis [29]. Although further experimental validation is still required, it was hypothesized that calmodulin interaction and KRas phosphorylation could modulate the localization of KRas in different discrete plasma membrane microdomains. In this model, phosphorylated KRas would possibly be enriched in microdomains containing specific effectors. Our follow-up studies indeed demonstrated by density–gradient fractionation and immunoelectron microscopy that Ser181-phosphorylated oncogenic KRas (KRasG12V) localized to nanoclusters at the plasma membrane that were distinct from the non-phosphorylated KRasG12V-S181A mutant [121]. Both PKC-activated oncogenic KRasG12V as well as the phosphomimetic KRas mutant (KRasG12V-S181D) co-clustered in the same plasma membrane nanodomains that also contained activated PI3K and Raf1. In PKC-inhibited cells, non-phosphorylated oncogenic K-Ras co-clustered with the KRasG12V-S181A mutant into nanodomains where PI3K was absent [121]. Therefore, PKC-mediated and calmodulin-sensitive Ser181 phophorylation permits segregation of KRas into specific plasma membrane nanodomains that favor an increased PI3K and Raf1 activation (Figure 3).

#### 3.4.4. Alternative Models for Calmodulin-Dependent KRas Signalling

Based on the work from Liao and coworkers [165], another model has been proposed [114,166]. This model favors the assembly of a ternary complex consisting of KRas, calmodulin, and PI3K at the plasma membrane to promote Akt signalling. Together with elevated Ca^2+^ levels observed clinically, and upregulated calmodulin levels found in many adenocarcinomas, this could support a key role for calmodulin in cancer initiation and progression. Although a ternary KRas/calmodulin/PI3K complex is plausible, experimental evidence for such a complex is still lacking. Additionally, the ability of the KRasG12V-S181A mutant to bind calmodulin but not PI3K, and the KRasG12V-S181D mutant to be defective in calmodulin, but not PI3K interaction (see above), cannot be easily reconciled. It should be noted that recent in vitro studies, using a phosphomimetic KRas S181E mutant, did not support a model of KRas S181 phosphorylation preventing calmodulin binding [83]. More studies, ideally in live cells, are needed to clarify this regulatory aspect of KRas/calmodulin interaction.

Additional support for Ca^2+^/calmodulin contributing to KRas-driven tumorigenicity has been provided. Wang and coworkers showed that binding of oncogenic KRas to calmodulin sequestered the number of calmodulin molecules available to activate CaMKII. This lack of CaMKII activation caused the suppression of non-canonical Wnt/Ca^2+^ signalling and triggered β-catenin activation that contributes strongly to tumourigenic properties of oncogenic KRas [167]. Moreover, inhibiting KRas/calmodulin interaction by stimulating PKC-mediated Ser181 phosphorylation of KRas, using the atypical PKC activator prostatin, suppressed tumourigenesis in KRas mutant pancreatic cancer cells as a result of increased CaMKII activity [167].

These results appear somewhat contradictory to those listed above that suggest PKC-mediated KRas phosphorylation to promote oncogenic potential. Different experimental conditions probably need to be considered to explain these discrepancies. For instance, the PKC activator prostratin [167] does not act as a tumour-promoting agent like more commonly used phorbol esters and indirect effects of this agent cannot be excluded. PKC activation stimulates cell survival and proliferation in cells expressing KRasG12V [168]. This is in line with PKC to function as a critical anti-apoptotic signal transducer in cells expressing activated KRas, promoting cell survival through the PI3K/AKT pathway [169,170]. Moreover, PKC inhibition reduced KRasG12V-induced tumour growth in vivo [168].

This highlights the difficulty to compare results obtained from the variety of tools available to modify PKC activity. Yet, the relevance of Ser181 phosphorylation of KRas in tumourigenesis was further demonstrated in pancreatic ductal adenocarcinoma (PDAC) in vivo. In KRas-induced tumours from PDAC cells, Ser181 phosphorylation was critical for KRas to interact with the heterogeneous nuclear ribonucleoproteins A2 and B1, which was required for PI3K/AKT activation, cell survival and tumour formation [171]. Likewise, Ser181 phosphorylation of oncogenic KRas was involved in tumour growth after subcutaneous injection of NIH3T3 fibroblasts or the colorectal cancer cell line DLD1 stably expressing phosphomimetic and non-phosphorylatable KRas mutants in nude mice [172]. Furthermore, treatment with pharmacological PKC inhibitors impaired tumour growth and correlated with KRas dephosphorylation and subsequent apoptosis [172]. Therefore, a substantial number of studies implicate that inhibition of PKC-mediated phosphorylation of KRas at the Ser181 residue, possibly in combination with agents that block calmodulin-mediated PI3K/Akt activation, could be a potential therapeutic target for KRAS-driven tumours (Figure 3).

## 4. The Multiple Modes of Calmodulin to Influence Rac1 GTPase Activity

### 4.1. Overview

Rac1 (Ras-related C3 botulinum toxin substrate 1) is a member of the Rho subfamily (Rho, Cdc42, Rac) [173,174] that controls cytoskeleton dynamics. Rho is responsible for stress fibers, while Cdc42 and Rac1 coordinate filopodia and lamellipodia formation, respectively. By controlling rapid actin re-arrangements, Rho-GTPases govern adhesion, migration, endocytosis, vesicular trafficking and proliferation [175,176,177,178,179]. Rac1 is a ubiquitous isoform of the Rac family that contains two other members. This includes Rac2, which is mainly expressed in the hematopoietic lineage, and Rac3, which is only found in neurons of the central nervous system [174,180]. Rac1 is the best studied member and its activity is critical for the proper functioning of various cellular processes such as pinocytosis, phagocytosis, exocytosis, cell–cell contacts, migration, axonal growth, differentiation, but also gene expression [94,176,178,181,182,183,184,185]. Most of these cellular events are facilitated via active Rac1 interacting with effectors that control actin dynamics. p21-activated kinase (Pak) is the central Rac1 effector that influences actin polymerization at the plasma membrane. This initially requires Pak-mediated phosphorylation of LIM kinase and cortactin, and entails actin-related protein 2/3 (Arp2/3) complex, neuronal Wiskott–Aldrich Syndrome protein (N-WASP)/WASP-family verprolin-homologous protein (WAVE) proteins, among others [186,187,188,189,190,191]. A large list of additional Rac1-GTP effectors also participate in actin cytoskeleton dynamics and other cellular responses. Alike all other GTPases, Rho-GTPases cycle between an active (GTP-bound) and inactive (GDP-bound) state, which is regulated by GEFs and GAPs [173,185,192,193,194]. Similar to Rab proteins, Rho guanine nucleotide dissociation inhibitors (RhoGDIs) provide an additional layer of regulation and complexity for Rho GTPases including Rac1. RhoGDIs preferentially bind inactive Rac1-GDP, working as a chaperone to maintain Rac1 as a soluble and inactive complex in the cytosol. This interaction makes Rac1-GDP inaccessible for GEFs located at the plasma membrane or endomembranes, and prevents nucleotide replacement in the Rac1 GDP/GTP cycle [179,195,196,197,198,199]. Highlighting the critical contribution of Ca^2+^ homeostasis in this regulatory circuit, elevated cellular Ca^2+^ levels following growth factor stimulation can activate PKC, which then phosphorylates RhoGDI. This promotes Rac1/RhoGDI dissociation, enabling the activation of Rac1 by GEFs associated with membranes [200].

#### 4.1.1. GEF-Mediated Rac1 Activation

Several GEFs specifically activate Rac1 and include Son of Sevenless 1/2 (Sos1/2), PAK-interacting exchange factor α/β (α/β-PIX), kalirin, T-lymphoma invasion and metastasis-inducing protein 1 (Tiam1) and Ras guanine nucleotide releasing factor (RasGRF). Other GEFs also activate Cdc42 or Rho such as Rho guanine nucleotide exchange factor 4 (ARHGEF4/Asef), Active breakpoint cluster region-related protein (Abr), Vav2/3, and Trio. Only few of those GEFs mentioned above have dual activity for Rac1 and Ras GTPases (Sos1 and RasGRF), and Tiam1 can act as a downstream effector of Ras [201]. Typical Rac1-specific GEFs contain tandem of diffuse B-cell lymphoma (DH) [202] and pleckstrin homology (PH) domains. The DH domain entails the GEF activity and the PH domain the phosphoinositode binding capacity with the highest affinity for phosphatidylinositol-3,4,5-triphosphate (PI(3,4,5)P_3_) [201,203]. Atypical GEFs, called dedicator of cytokinesis protein superfamily proteins [204], lacking DH or PH domains, have also been described [205].

Although many GEFs that activate Rac1 have been reported, each one differentially modulates Rac1-regulated events. This may suggest that spatiotemporal regulation of compartmentalized GEF activity and tissue specificity ensure the final outcomes for Rac1 activity in different cell types. All these GEF proteins are activated by signalling cascades, usually initiated by kinases activated by plasma membrane receptors. GEF activation may occur through different or combinatory modes: exchanging their subcellular localization, releasing their auto-inhibitory conformation and/or by posttranslational modifications, such as serine/threonine (Ser/Thr) or tyrosine (Tyr) phosphorylation [201]. For instance, Rac1 can traffic from the plasma membrane to early and late endosomes by vesicular transport, to be activated after membrane-associated delivery into the vicinity of endosome-associated GEFs Tiam1 or Vav2 [94,106,206,207,208]. Overall, a substantial number of reports demonstrate that calmodulin and CaMBPs modulate many Rac1-related GEF activities. More detailed information supporting prominent functions for calmodulin in GEF-mediated regulation of Rac1 signalling and effector interactions is discussed in the following section (Section 4.3).

#### 4.1.2. Rac1 Membrane Association

Strikingly different from the mode of action of calmodulin on KRas (see Section 2), calmodulin/Rac1 complex formation does not affect dynamics of Rac1 membrane association. Based on results from our group examining Rac1 and KRas endosomal membrane association, the higher membrane affinity of Rac1 compared to KRas might substantially contribute to different Rac1 and KRas membrane dynamics [100]. Rac1 is inserted into membranes by a geranyl-geranyl (20-carbons isoprenyl), and possibly by an additional palmitoyl (16-carbons) group [209], whereas membrane anchoring of KRas only requires a farnesyl (15-carbons) group. In the case of Rac1, after its synthesis in the cytosol as a hydrophilic protein, the leucine residues in the C-terminal CAAX box (CLLL) allow complex formation with geranyl-geranyl transferase type I to link the geranyl-geranyl group to the cysteine residue. This modification enables Rac1 recruitment to ER membranes, where Rce1 and Icmt enzymes (see Section 3) further modify Rac1. After these posttranslational modifications, Rac1 is extracted and solubilized from ER membranes to the cytosol by RhoGDIs, which confers an additional level of regulation compared to KRas [179,196]. In the following section, the domains within calmodulin that interact and influence Rac1 activity will be discussed.

### 4.2. The Consequences of Direct Rac1/Calmodulin Interactions

Over the years, it has become clear that calmodulin not only regulates Rac1 function via direct protein–protein interactions, but also indirectly through the diverse action of other CaMBPs. These studies reveal a complex molecular crosstalk and interplay of calmodulin with Rac1 and its effectors and regulators. This seems to enable calmodulin to decode signals derived from changes in Ca^2+^ homeostasis into cytoskeleton re-arrangements that control cell adhesion, morphology, and migration [210].

#### 4.2.1. Calmodulin Interacts with Active Rac1 in a Ca^2+^-Dependent Manner

Pulldown and co-immunoprecipitation assays using endogenous or ectopically expressed wild-type and mutant versions of Rac1 enabled several groups to thoroughly characterize the interaction of Rac1, but also Cdc42, with calmodulin and its consequences for signal outcome [21,22,23,211,212]. Overall, evidence points at a Ca^2+^-dependent and direct interaction of endogenous and GTP-bound (active) Rac1 and Cdc42 with calmodulin [22,23]. Alike KRas, Rac1 contains several motifs that participate in calmodulin binding but only some of them are essential (Figure 4).

Based on the screening of various databases for potential calmodulin-binding domains [7] (http://calcium.uhnres.utoronto.ca/ctdb) and in vitro peptide competition assays, Bhullar and coworkers identified a C-terminal 14 amino acid region (residues 151-164) of Rac1 important for calmodulin binding [23]. Within that region, the positively charged lysine 153 (Lys153) and arginine 163 (Arg163) residues are critical for calmodulin interaction [21]. Replacement of these two residues with alanine in a double mutant (K153A/R163A) strongly diminishes EGF-induced Rac1 activation [21]. Thus, based on these interaction studies, direct binding of calmodulin positively regulates Rac1 activity (Figure 4).

The Rac1 C-terminus (amino acids 170-192) containing the PBR and geranyl-geranyl lipid anchor binds calmodulin albeit with a lower affinity than full length Rac1. In fact, calmodulin binding is abrogated in a Rac1 mutant lacking the PBR or the prenyl group. This indicates that both regions are essential for interaction with calmodulin [22] (Figure 4).

Several Rac1 effectors also interact with the C-terminal Rac1 domain, including phosphatidylinositol-4-phosphate 5-kinase (PIP5K) [213,214,215]. Studies from our group identified that calmodulin competes with PIP5K for Rac1 binding, which has important implications for the efficiency of Rac1 to promote PIP5K-mediated production of phosphatidylinositol-4,5-biphosphate (PI(4,5)P_2_) [22]. At the plasma membrane, generation of PI(4,5)P_2_ can regulate actin dynamics to control migration and endocytosis [216,217]. Accordingly, inhibition of calmodulin with W13 increases Rac1 binding to PIP5K. This subsequently elevates PI(4,5)P_2_ levels, leading to plasma membrane reorganizations that affect the clathrin-independent endocytic pathway. Hence, these W13-induced changes alter clathrin-independent internalization of cargos like major histocompatibility complex 1, cholera toxin and β1-integrins. Ligands that follow the clathrin-dependent internalization pathway, such as transferrin or polymeric immunoglobulin A, are not affected [22,212,218].

Inhibition of calmodulin using W13, or its analog W7, which rapidly diffuse into cells, have been proven powerful tools to address how calmodulin influences the spatiotemporal regulation of particular pathways [219]. Analysis of these small molecule inhibitors not only allowed validation of calmodulin stimulating growth factor-induced Rac1 activation, but also revealed calmodulin contributing to control basal Rac1 activity. W13 and analogs also greatly contributed to better characterize Rac1 activation by different stimuli in various cell types and conditions [21,22,23,157,158,211,220,221,222,223]. However, the underlying molecular mechanism how complex formation between calmodulin and Rac1 alters Rac1 activity and signal output is still not fully understood. As mentioned above, calmodulin might function as a GEF for Rac1 [21]. Alternatively, calmodulin may inhibit the access or activity of one or multiple Rac1-specific GAPs, although experimental evidence for this is lacking. Other more plausible possibilities include the calmodulin-dependent indirect regulation of Rac1 activity via IQGAPs or several Rac1-GEFs, which will be discussed in Section 4.2.2 and Section 4.3.

#### 4.2.2. Calmodulin Modulates the Interaction of IQGAP with Rac1 to Facilitate Cytoskeleton Rearrangements

IQGAP is a multidomain scaffolding protein that directly interacts with Rac1 or Cdc42 and F-actin to modulate actin dynamics at the cell surface to control cell morphology, cell–cell adhesion and motility [224,225,226]. These interactions occur via the C-terminal region of IQGAP that contains a RasGAP-related domain (GRD) and an adjacent Ras-GAP domain that also has high affinity for both GTPases [224,227] (Figure 4). These features probably contribute to specificity, as IQGAPs only interact with Rac1 and Cdc42, but not RhoA or Ras proteins [228,229]. Biochemical assays demonstrate exclusive binding of IQGAP1 to active Rac1-GTP and Cdc42-GTP, but not inactive Rac1/Cdc42-GDP [228]. This correlates with the presence of IQGAP1 at membrane ruffling areas and cell–cell contacts of epithelial cells [228]. The IQGAP2 isoform, which is predominantly expressed in the liver, also interacts with Rac1 and Cdc42 [229]. In contrast to IQGAP1, this interaction appears independent of the activation status (GDP or GTP) of these GTPases. Mechanistically, both IQGAPs contain a GRD domain that does not exert GAP activity towards Rac1 or Cdc42, but rather inhibits GTP hydrolysis, thereby potentiating and stabilizing their active conformation. Hence, complex formation of active Rac1 and Cdc42 with IQGAPs significantly increases and prolongs actin polymerization at the leading edge of migrating cells [51,210,230,231,232,233,234]. This activity critically drives cell motility, has been linked to migration and invasion in cancer metastasis, and has been reviewed in detail elsewhere [232].

Calmodulin binds IQGAP, yet this interaction is complex. The N-terminal calponin homology domain (CHD) in IQGAP can bind Ca^2+^/calmodulin and F-actin in a competitive, but mutually exclusive manner [48,49,235,236,237]. In addition, IQGAP contains four IQ motifs (IQ1-4), which are commonly considered to bind calmodulin even in the absence of Ca^2+^ (apocalmodulin) [3]. Interestingly, although all four IQ motifs in IQGAP1 can bind Ca^2+^/calmodulin, only two of those IQ motifs (IQ3-4) bind apocalmodulin [238,239]. The potential function of the latter interaction remains to be clarified, as apocalmodulin differs greatly from calmodulin, does not require Ca^2+^ and binds a plethora of other proteins [238].

As described above, calmodulin competes with filamentous actin (F-actin) for the binding to IQGAP1 in a mutually exclusive manner, as evidenced by the purification of two pools of adrenal IQGAP that either contained F-actin or calmodulin [237]. IQGAP1 may stimulate cell migration through two interactions with F-actin. While the N-terminal CHD domain bundles F-actin, the C-terminal domain of IQGAP caps barbed ends of F-actin. Both of these interactions are inhibited by calmodulin in the presence or absence of Ca^2+^, possibly keeping IQGAP1 in a closed conformation [234]. Similar competitive interactions between IQGAP1 and Ca^2+^/calmodulin for a variety of others IQGAP1 effectors and signalling proteins have been identified. The underlying mechanism are not fully resolved and some results suggest that Ca^2+^ binding to calmodulin promotes its dissociation from IQGAP1. This could enhance interactions of IQGAP with effectors responsible for cytoskeleton organization [240]. On the other hand, it appears more likely that Ca^2+^/calmodulin may dissociate active Rac1 or Cdc42, beta-catenin and E-cadherin from IQGAP1 to affect actin organization and cell–cell adhesion in epithelial cells [224,232,241]. Taking into account that IQGAP1 promotes cell migration by interacting and stabilizing active Rac1 and that Ca^2+^/calmodulin dissociates Rac1 from IQGAP1, expression of a IQGAP1 mutant defective in Ca^2+^/calmodulin binding enhances migration in MCF-7 cells [231]. Therefore, inhibition of calmodulin may upregulate the multiple functions related to cell motility that depend on IQGAP1-Rac1/Cdc42 interactions [49,224,226,241]. This role of calmodulin, inhibiting the ability of IQGAP1 to interact with Rac1 and/or Cdc42, has also been shown critical for enteropathogenic *Escherichia coli* (EPEC) infection in HeLa cells [242]. In these studies, EPEC-induced IQGAP1/calmodulin interaction leads to the dissociation of Rac1 and Cdc42 from IQGAP1. This then triggers an actin reorganization at the plasma membrane to promote an actin pedestal formation necessary for EPEC infection.

Another hypothetical scenario proposes calmodulin to participate as an accessory scaffolding protein to assemble Rac1 with IQGAP, but not Cdc42 [224]. However, this model still remains to be validated experimentally in cell culture models and to date the published data rather supports Ca^2+^/calmodulin binding to IQGAP1 to favor the dissociation of several partners including Rac1 and Cdc42.

### 4.3. Calmodulin Influences the Activation of Several Rac1-specific GEFs

Besides the direct interaction of calmodulin with Rac1 (see Section 4.2.1) and its control over IQGAP-dependent Rac1 activation (see Section 4.2.2), calmodulin, and calmodulin-binding effectors can activate Rac1 through several Rac1-GEFs, which are summarized above (Figure 4).

Rac1-GEFs commonly contain a PH domain (see Section 4.1) that allows interaction with phosphoinositides, facilitating Rac1-GEF recruitment to the plasma membrane or endomembranes, and their subsequent activation. Several Rac1-GEFs (Tiam1, Vav, and others) are modulated by PI3K, since its product, PI(3,4,5)P_3_, can bind to their PH domains [243,244,245]. This regulatory circuit is relevant for a large number of Rac1-GEFs and has been reviewed in great detail by others [205]. Ca^2+^/calmodulin has the capacity to bind and activate PI3K [143,159,160,161]. Hence, Ca^2+^/calmodulin may activate Rac1-GEFs and consequently increase Rac1-GTP loading through upregulation of PI3K activity (Figure 4). Indeed, in activated neutrophils, calmodulin, and PI3K inhibitors cooperate to downregulate Rac1 activity [220].

Several CaMBPs, in particular kinases, also modulate Rac1-GEF activity and membrane translocation via phosphorylation events. Several Rac1-GEFs are phosphorylated by different members of the diverse CaMK family, including CaMKI, CaMKII, CaMKIV, and CaMK kinase (CaMKK), all of which being activated upon Ca^2+^/calmodulin binding that relieves their auto-inhibited conformation [246,247].

A number of studies established that in neurons and other cell types, cytosolic Ca^2+^ elevation stimulates the Rac1-GEFs, Tiam1, Kalirin, and β-Pix, in a CaMKII-dependent manner [222,248,249,250,251,252,253]. Using NIH3T3 and Swiss 3T3 fibroblasts, Exton and coworkers identified that platelet-derived growth factor activated and translocated Tiam1 to membrane fractions by CaMKII-mediated threonine phosphorylation that was coupled to phospholipase C (PLC)-driven cytosolic Ca^2+^ increase [157,158,223].

In post-synaptic neurons, Ca^2+^ entry pulses induced by glutamate binding to the N-methyl-D-aspartate (NMDA) receptor, produce a reciprocal and synergistic activation of CaMKII and Tiam1 to generate persistent Rac1 activation. This then ensures stable actin polymerization to maintain spinal structure during long-term potentiation [249]. This is in line with earlier studies implicating Tiam1 to be essential for Rac1-dependent actin remodeling during NMDA receptor-dependent regulation of spine development [253]. In addition, Ca^2+^/calmodulin-dependent CaMKII phosphorylated and inactivated RhoGAP is involved in beta-catenin/N-cadherin and NMDA receptor signalling, thereby increasing Rac1-GTP levels required for dendritic spine morphology [254].

The NMDA receptor also stimulates a signalosome complex containing CaMKK, CaMKI, β-PIX and G-protein-coupled receptor (GPCR)-kinase-interacting proteins (GIT) to promote spinogenesis and synaptogenesis in cultured neurons and hippocampal slices. In these models mimicking spine development, Ca^2+^/calmodulin triggers CaMKI-mediated phosphorylation of Ser516 in β-PIX, which stimulates its GEF activity towards Rac1, driving PAK-dependent spinogenesis [250]. These complex mechanisms are not only relevant for ion channel receptors in neuronal cells, but the same CaMKK/CaMKI/β-PIX/GIT axis also contribute to Rac1 activation required for estrogen-inducible medulloblastoma cell migration [251].

On the other hand, neurotransmitters such as glutamate can activate Ca^2+^-permeable receptors, including the NMDA receptor. This ultimately stimulates Ca^2+^/calmodulin binding to RasGRF through IQ motifs, which leads to elevated Rac1-GTP levels to then trigger MAPK signal cascade activation and control neuronal synaptic plasticity [137].

Yet, besides the multiple modes of Ca^2+^/calmodulin influencing Rac1 activity, it should be noted that calmodulin can also act downstream of Rac1 to control actin dynamics through the regulation of myosin light chain kinase (MLCK) [255]. In this study, phorbol ester-activated PKCδ, through the up-regulation of the Tiam1/Rac1/calmodulin/MLC2/MLCK cascade, elicited actomyosin formation in the protruding leading edge of migratory human peripheral blood T-lymphocytes [255,256].

Finally, transamidation is a post-translational modification that only recently was identified to affect Rac1 activity. In the A1A1v rat embryonic cortical cell line, stimulation of serotonin 2A receptor increases Rac1 transamidation and activity [257], which is coupled to PLC-dependent Ca^2+^/calmodulin signalling [258]. Ca^2+^/calmodulin activates transglutaminase, which participates in the addition of amines (e.g., serotonylation), crosslinking of protein-bound glutamine (Gln) or deamination of Gln residues of small GTPases. The fact that stimulation of serotonin 2A receptors in primary cortical neurons, as well as muscarinic acetylcholine receptors or NMDA receptors in neuronal cell lines, results in trans- or deamidation of Rac1 at Gln61, could suggest that Ca^2+^/calmodulin-induced transamidation activates Rac1 by inhibiting its GTP hydrolysis.

In summary, based on currently available literature summarized in the previous sections, it appears that depending on the microenvironmental changes, cells have an enormous variety of mechanisms at their disposal to allow Ca^2+^/calmodulin-dependent changes in Rac1 activity to alter a diverse set of cellular activities (Figure 5).

## 5. Adding Another Layer of Complexity: Rac1 is a KRas Effector

In the previous sections we summarized the extremely complex regulatory network that enables calmodulin to modulate the location, kinetics and effector pathways driven by two small GTPases, KRas and Rac1. It is important to note that current methodologies are still limited to identify a spatiotemporal hierarchy amongst calmodulin-activated signalling cascades. This has made it difficult to determine primary, secondary, and less prominent targets regulated by Ca^2+^/calmodulin in a ligand- and cell-type specific manner. It is tempting to speculate that a substantial number of effector signals promoted by KRas and Rac1 result from their direct interaction with calmodulin. In addition, Ca^2+^/calmodulin interacts with a plethora of Rac1 regulators and other CaMBPs to indirectly control Rac1 signal outcome (see Section 4.2, Section 4.2.1, Section 4.2.2 and Section 4.3).

Further complexity is added by Rac1 being an effector of Ras signalling via PI3K-dependent and -independent pathways that activate Rac1-GEFs [205,244,245,253,259,260,261,262,263,264,265]. The most prominent of these routes, with remarkable significance for tumour growth and progression, is represented by Ras-GTP interacting with PI3K, which can then activate Rac1. Firstly, disruption of Ras/PI3K interaction leads to inhibition of Rac1 activation and tumour regression in a mouse model for EGFR mutant lung adenocarcinoma [266]. Secondly, Rac1 activity is required in HRas-induced tumourigenesis [24,266,267,268,269]. Thirdly, loss of Rac1 causes a substantial reduction in a mouse model of KRas mutant-driven pancreatic, epidermal papilloma, and lung tumour growth [25,270,271].

In addition to Rac1-dependent stimulation of proliferation and cell migration (see Section 4), Rac1-mediated anti-apoptotic signals also favor Ras-dependent oncogenicity [272]. Pak1 is considered the main Rac1 effector that influences cortical actin cytoskeleton rearrangements most relevant for cell motility (see Section 4.1). Rac1-mediated Pak1 activation also appears fundamental in KRas-driven skin tumour progression through its decisive role in MAPK and Akt activation [273]. In this setting, Pak1 stimulates Raf1 kinase Ser338 phosphorylation, which increases MAPK activity but also upregulates the PI3K/Akt pathway. The latter signalling cascade is further promoted through a cooperative action of Rac1 with others Rho-GTPases, leading to PI3K activation [274,275].

The events triggered by Ras oncogenes and facilitated by Rac1 and its effectors point at the therapeutic potential of targeting Ca^2+^/calmodulin in tumour growth and metastasis. The precise roles of calmodulin in Ras-driven tumourigenesis are not fully understood, but Ca^2+^/calmodulin-induced Rac1 activation may strengthen tumour growth as well as metastasis. As proof-of-principle, calmodulin inhibitors abrogate PI3K and Rac1 activation [22,23,143,159,220]. This concept may indicate some promise for therapeutic anti-cancer approaches, yet aiming to block calmodulin function may only be suitable for some cancers. The interaction of Ca^2+^/calmodulin with active KRas impairs PKC-mediated KRas Ser181 phosphorylation, which is considered to increase oncogenic potential of KRas [29,172] (for further details see also Section 3) (Figure 3). However, evidence for an involvement of calmodulin in KRas-driven tumourigenesis is still lacking and further studies are needed to clarify its contribution in de-regulated and KRas-dependent oncogenic events.

Finally, one should also consider that calmodulin could couple Ras and Rac1 signalling pathways by regulating GEFs with affinity towards both GTPases [205,276]. For example (see also Section 4.1.1 and Section 4.3), Ca^2+^/calmodulin binding to RasGRF activates Ras, Rac1, and effectors, including MAPK, in response to signals from a variety of neurotransmitters [137]. On the other hand, earlier studies [245,262] suggest that Ca^2+^/calmodulin, most likely via PI3K, could enable Sos1 to simultaneously activate Ras and Rac1, hence synergizing signal outcome of both GTPases.

Taken together, besides the direct and indirect modes of action that allow Ca^2+^/calmodulin to influence KRas and Rac1 activity, Rac1 can also act as an effector of Ras signalling cascades. This connectivity between Ras and Rac1 signalling is often observed in oncogenic settings, which are characterized by continuously active mutant forms of growth factor receptors or Ras isoforms. However, Ca^2+^/calmodulin-mediated changes in KRas and Rac1 signalling in normal cells are based on the transient nature of environmental signals that trigger fluctuations in Ca^2+^ levels. Future research therefore needs to clarify how the de-regulation of Ca^2+^ homeostasis in tumour cells alters the availability of ‘active’ calmodulin and how this might influence the performance of small molecules targeting Ca^2+^/calmodulin.

## 6. Perspectives

Over the years, taking advantage of the rapidly improving technologies available, the field has made substantial progress to dissect the highly versatile and complex regulatory circuits that allow calmodulin to influence signalling pathways. This has provided the field with a range of therapeutic targets, including several calmodulin-binding proteins, such as CaMKs. To summarize the enormous variety of Ca^2+^/calmodulin-dependent mechanisms that cells have at their disposal, some of those with therapeutic potential, would have gone beyond the scope of this review. We therefore refer the reader to recent excellent reviews that cover other aspects of calmodulin in cellular pathophysiology. In this article, we focused to provide an overview on the complex network of calmodulin interactions that influence the spatiotemporal signalling of KRas and Rac1 GTPases, with consequences for biological outcomes in health and disease.

Calmodulin not only directly interacts with KRas and Rac1 to influence their localization and activity, but through other CaMBPs, also modifies the capacity of several GEFs and GAPs to control the GTP/GDP cycle of both GTPases. These multifaceted communications with KRas and Rac1 GTPases triggered by calmodulin have manifold consequences for effector pathways that shape the ability of cells to proliferate, migrate, differentiate or respond to death signals.

Some of these diverse mechanisms for calmodulin to regulate KRas and Rac1 GTPases could provide therapeutic opportunity in cancer pathophysiology, in particular where aberrant KRas signalling drives Rac1-dependent activities that contribute to tumour progression. Given the high incidence of oncogenic KRas mutations in human cancers, and the ability of calmodulin to modulate oncogenic KRas activity, targeting calmodulin in these settings may unravel novel treatment options. Future studies aiming to substantiate a role for calmodulin in KRas-driven tumourigenesis will be required to clarify its therapeutic potential. Despite this concept having therapeutic anti-cancer promise, one also needs to consider that calmodulin is ubiquitously expressed. Hence, avoiding unspecific side-effects and circumventing de-regulation of Ca^2+^ homeostasis and Ca^2+^/calmodulin-responsive signalling cascades in neighbouring healthy cells and tissues will be critical. Thus, besides the efforts to advance calmodulin-based anti-cancer techniques, future research will also have to establish drug carriers or platforms that can ensure localized delivery of pharmaceutical agents to specifically block calmodulin-dependent oncogenic signalling in localized tumours.

## Figures and Tables

**Figure 1 ijms-21-03680-f001:**
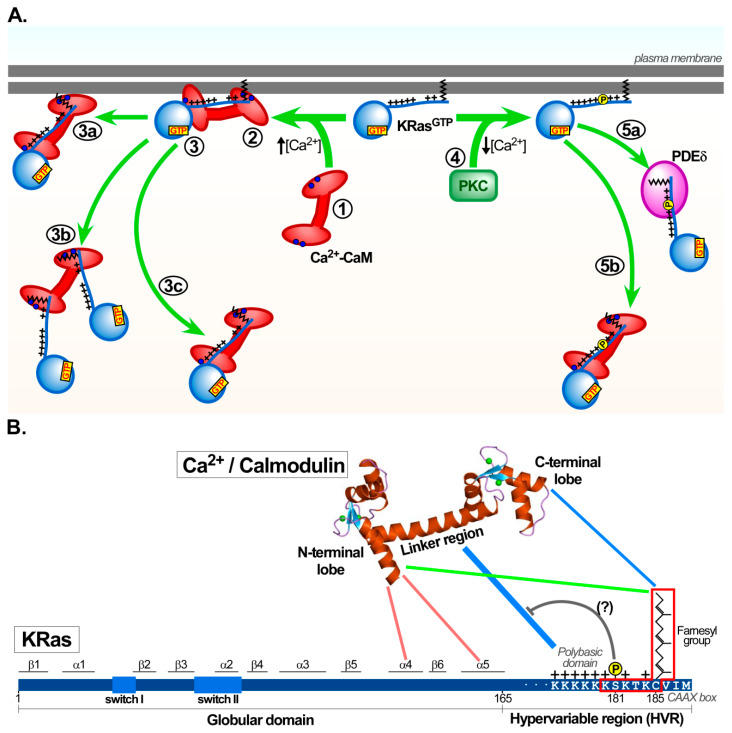
**Calmodulin/KRas interaction motifs and their role in KRas trafficking.** The different possibilities how calmodulin may affect KRas trafficking (A) and the protein domains facilitating interaction of calmodulin and KRas are shown (B). (**A**) Role of calmodulin in KRas trafficking routes. (**1**) Upon elevation of intracellular [Ca^2+^] levels, Ca^2+^ binds and activates calmodulin (Ca^2+^-CaM), which preferentially interacts with active KRas (KRas-GTP) at the plasma membrane. (**2**) Initially, the C-terminal lobe and the linker region of Ca^2+^-CaM interact with the polybasic domain of KRas. The negatively charged Ca^2+^-CaM linker region is attracted to the polybasic KRas domain through electrostatic coupling. This may be followed by a conformational change in both proteins that permit a secondary interaction between the N-terminal lobe of Ca^2+^-CaM with the globular domain of KRas, in particular with helices α4 and α5. (**3**) The majority of Ca^2+^-CaM/KRas complexes remain associated with the plasma membrane. However, to some extent, the hydrophobic C-terminal lobe of Ca^2+^-CaM interacts and extracts the KRas farnesyl group from the lipid bilayer. (**3a**) KRas could remain at the plasma membrane through the interaction of Ca^2+^-CaM with the acidic membrane leaflet or through interaction with other proteins that are recruited to the plasma membrane, such as phosphoinositide 3-kinase (PI3K). (**3b**) Alternatively, after binding of the C-terminal lobe of Ca^2+^-CaM to KRas, followed by a conformational change in Ca^2+^-CaM, a second KRas protein may bind to the N-terminal lobe of Ca^2+^-CaM. This would result in a 2:1 Ca^2+^-CaM/KRas stoichiometry. These KRas proteins could then be removed from the plasma membrane after interaction with Ca^2+^-CaM. (**3c**) Ca^2+^-CaM/KRas complexes with 1:1 stoichiometry may also detach from the plasma membrane. (**4**) Reduced cytosolic Ca^2+^ levels lead to the dissociation of the CaM/KRas complex, enabling protein kinase C (PKC) to phosphorylate active KRas at the serine 181 residue (Ser181). (**5a**) PDEδ or (**5b**) under certain conditions Ca^2+^-CaM could cause KRas dissociation from the plasma membrane. The pool of KRas proteins trafficking along the various routes is likely to vary in different cell types and be influenced by changes in the microenvironment and physiological conditions. See text for further details (Section 3.3). (**B**) Regions involved in KRas and calmodulin interaction. The polybasic domain and the farnesyl group within the hypervariable region (HVR) of KRas are both essential to interact with the linker region and the C-terminal lobe of Ca^2+^/Calmodulin (blue lines), respectively. The KRas polybasic domain is the initial and primary binding region that interacts with Ca^2+^/calmodulin (thicker blue line). The minimal Ca^2+^/calmodulin binding motif of KRas, KSKTKC-farnesyl, is indicated (red square). As part of the KRas/calmodulin interaction, helices α4 and α5 of the globular domain of KRas could also participate interacting with the N-terminal lobe of calmodulin (stoichiometry 1:1; red lines). Alternatively, KRas may first interact with the C-terminal lobe of Ca^2+/^calmodulin. This could trigger a conformational change of calmodulin to allow interaction of a second KRas protein to the calmodulin N-terminal lobe (stoichiometry 2:1; green line). The position of α-helices, switch I and II within the globular domain (1–165 aas) and the HVR (166–188 aa) of KRas are indicated. The calmodulin N- and the C-terminal lobes, the linker region and the two bound Ca^2+^ ions in each lobe are also depicted.

**Figure 2 ijms-21-03680-f002:**
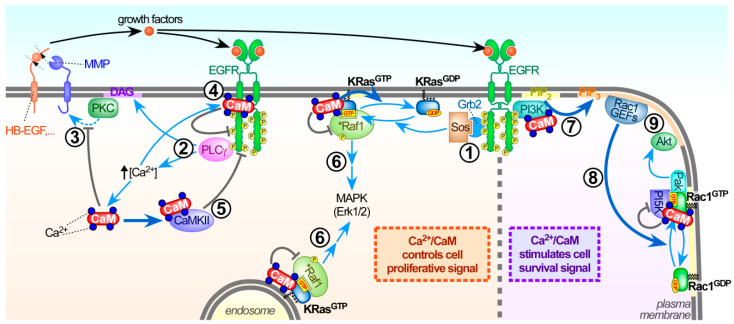
**Model of key signalling events modulated by Ca^2+^/calmodulin and decisive for cell proliferation and survival.** In this scheme, key signalling events regulated by calmodulin (CaM) and critical for cell proliferation (left) and cell survival (right) are highlighted. **Left:** Several protein–protein interactions of calmodulin enable a negative feedback regulation of the mitogen-activated protein kinase (MAPK) extracellular signal-regulated kinase 1/2 (Erk1/2). This signalling cascade is initially activated by epidermal growth factor receptor (EGFR), and other receptor tyrosine kinases not shown here, at the cell surface and prolonged on endosomes. The overall outcome of these interactions prevents a strong and sustained signal that promotes cell proliferation. **Right:** Calmodulin can activate phosphoinositide-3 kinase (PI3K) and through protein kinase B (Akt) and Rac1, ensure the generation of signals that stimulate cell survival. Key interactions and signalling events are numbered and underscore the following: (**1**) Growth factor (e.g., EGF) ¡-induced EGFR activation and autophosphorylation enables the recruitment of adaptors (Grb2) and guanine nucleotide exchange factors (Sos). This is followed by activation of the Ras/Raf1/MAPK signalling pathway. (**2**) Simultaneously, phospholipase C γ (PLCγ) is recruited to activated EGFR to produce the second messenger diacylglycerol (DAG) and elevate cytosolic Ca^2+^ levels. (**3**) The latter binds and activates calmodulin, which disables DAG-dependent and protein kinase C (PKC)-mediated activation of matrix metalloproteases (MMPs). This prevents the shedding and release of membrane-bound growth factor precursors, such as heparin-binding EGF (HB-EGF). This regulatory circuit effectively prevents prolonged EGFR activation in an autocrine manner. (**4**) In addition, Ca^2+^/calmodulin binds EGFR and inhibits its tyrosine-kinase activity. (**5**) Moreover, Ca^2+^/calmodulin binds Ca^2+^/calmodulin-dependent kinase II (CaMKII), which has the ability to phosphorylate and inactivate EGFR. These multiple interactions suggest a negative feedback mechanism that allows Ca^2+^-induced calmodulin activation to trigger downregulation of EGFR signalling. (**6**) Furthermore, Ca^2+^/calmodulin may preferentially interact with active KRas (KRas-GTP), which is particularly relevant in settings where KRas is the most prominent Ras family member contributing to MAPK activation, such as in NIH3T3 fibroblasts. In these cells, KRas/calmodulin interaction blocks downstream Raf1 activation. This could possibly be accompanied by the recruitment of GTPase activating proteins (GAPs) that further reduce KRas-GTP levels, thereby advancing to downregulate MAPK signalling and proliferation in these cells. (**7**) On the other hand, Ca^2+^/calmodulin together with activated EGFR can stimulate PI3K activity. (**8**) This increases the recruitment and activity of Rac1 guanine nucleotide exchange factors (Rac1 GEFs) that elevate Rac1-GTP (active) levels. (**9**) Together with Akt activation, these signalling events are linked to an anti-apoptotic response that triggers cell survival. The GTP/GDP cyles of KRas and Rac1, as well as the membrane location of phosphatidyl-4,5-biphosphate (PIP_2_) and phosphatidylinositol-3,4,5-triphosphate (PIP_3_) are indicated. In summary, Ca^2+^/calmodulin modulates the MAPK pathway driving proliferation and on the other hand, stimulates PI3K activity to induce cell survival. The overall biological outcome probably depends on the signal diversity derived from the extracellular milieu as well as the cell-specific and differential repertoire of calmodulin-responsive players in each cell.

**Figure 3 ijms-21-03680-f003:**
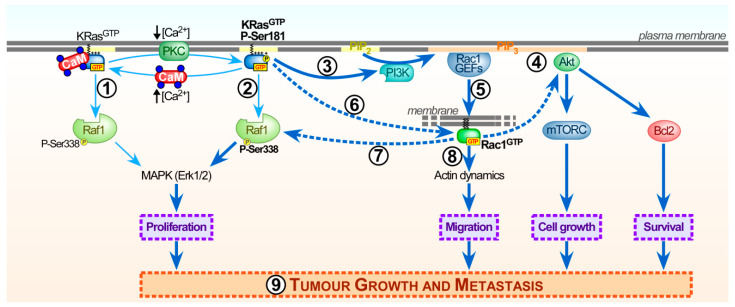
**Model of calmodulin and protein kinase C (PKC) modulating critical features of KRas-driven tumourigenesis**. This hypothetical model is based on studies comparing the impact of pharmacological compounds modulating PKC or calmodulin activity in a variety of cell models expressing wild-type or oncogenic and constitutively active KRas (KRasG12V) containing Ser181-phosphomimetic or non-phosphorylatable mutants (see Section 3.4.3 for further details) [16,29,105,121,169,170]. (**1**) Ca^2+/^calmodulin binding to the polybasic region (PBR) within the hypervariable region (HVR) of KRas inhibits PKC-mediated Ser181 phosphorylation of KRas (P-Ser181) by sterical hindrance. Complex formation of calmodulin with (active) KRas-GTP may segregate KRas to membrane microdomains where KRas-GTP could be more susceptible to GTPase activating protein (GAP)-mediated KRas inactivation. This ensures KRas inactivation and termination of KRas-GTP-mediated activation of phosphoinositide 3-kinase (PI3K) effector pathways. (**2**) Low Ca^2+^ levels disrupt KRas/calmodulin interaction and favor PKC-mediated Ser181 phosphorylation of KRas. (**3**) This allows a conformational change in P-Ser181 KRas and is followed by its segregation to distinct plasma membrane microdomains or endosomal membranes (omitted in this scheme), where interaction and activation of effectors like PI3K can occur. (**4**) Ser181 phosphorylation of oncogenic KRas (KRasG12V) triggers PI3K and Akt-dependent anti-apoptotic signals driven by B-cell lymphoma 2 (bcl2) and mammalian target of rapamycin complex (mTORC) that promote survival and cell growth, respectively. (**5**) PI3K also activates Rac1 guanine nucleotide exchange factors (Rac1 GEFs) that promote activation of Rac1 (Rac1-GTP) on plasma (or endosomal) membranes (dashed lines). The membrane location of phosphatidyl-4,5-biphosphate (PIP2) and phospharidylinositol-3,4,5-triphosphate (PIP3) is indicated. (**6**) Alternatively, active KRas (KRas-GTP) can directly associate with a Rac1-GEF to activate Rac1. (**7**) Vice versa, active Rac1-GTP and its effector Pak1 have been suggested to facilitate Ser338 Raf1 phosphorylation and activation, which affects proliferation along the Raf1/mitogen-activated protein kinase (MAPK) pathway. (**8**) Rac1-GTP drives actin dynamics linked to cell migration. (**9**) These complex regulatory networks are highlighted by the requirement of Rac1 activity in KRas-driven cancers (see Section 5 for further details). Overall, Ser181 phosphorylation of oncogenic KRas is at the forefront of multiple signalling pathways that are fundamental to cellular events that drive tumour growth and metastasis. This can be counteracted by KRas/calmodulin complex formation, providing a potential tool to reduce signal output of oncogenic KRas.

**Figure 4 ijms-21-03680-f004:**
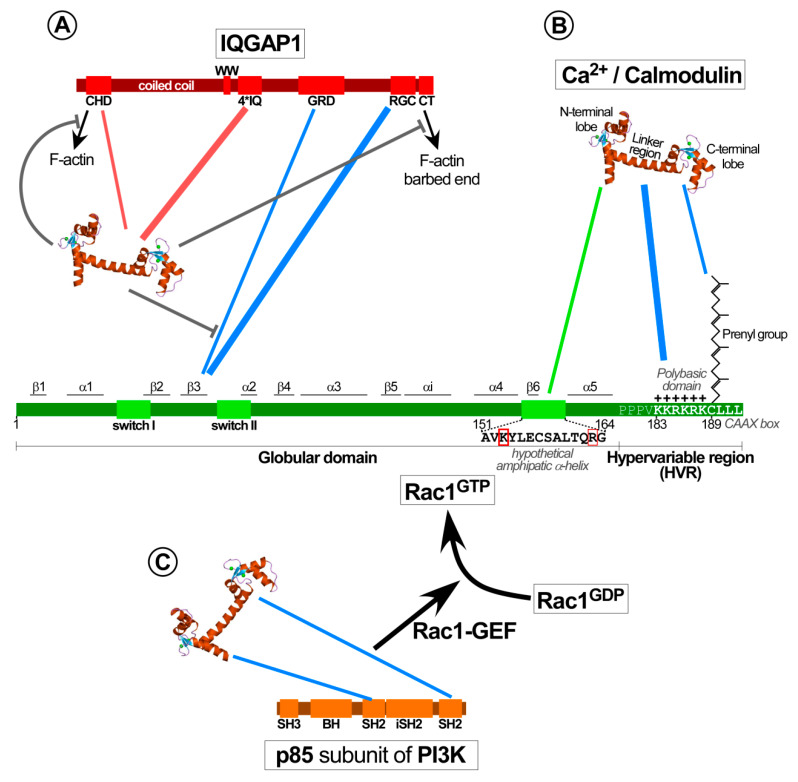
**Calmodulin regulates Rac1 signalling through direct and indirect protein–protein interactions.** The scheme summarizes the multiple interactions between calmodulin, Rac1, isoleucine–glutamine (IQ) Motif Containing GTPase Activating Protein 1 (IQGAP1) and phosphoinositide 3-kinase (PI3K). Ca^2+^/calmodulin affects Rac1 signalling outcome by directly interacting with Rac1 (A), modulating IQGAP1/Rac1 interaction (B) or activating PI3K (C). (**A**) Multiple Rac1 domains directly interact with Ca^2+^/calmodulin. Similar to the KRas/calmodulin interaction (see Section 3.3 and Figure 1), the polybasic domain (PBR) and the geranylgeranyl group (prenyl) within the hypervariable region (HVR) of Rac1 are both essential for Ca^2+^/calmodulin binding (blue lines). Based on the well-characterized KRas/calmodulin interaction, one can speculate that the linker domain and C-terminal lobe of calmodulin interact with the PBR and prenyl group of Rac1, respectively. In addition, amino acids 151-164 in Rac1 may adopt an amphipathic α-helix that contributes to calmodulin interaction (green line) and Rac1 activation. Within this region, the basic amino acid K153 (thick red square), and to a lesser extend R163 (red square) are critical for the interaction with the N-terminal lobe of calmodulin. The position of the switch I and II domains, the PBR (aa 183-188) and the prenyl group attached to C189 of Rac1 are indicated. The N- and the C-terminal lobes and the linker region of calmodulin as well as two bound Ca^2+^ ions in each lobe are also shown. (**B**) IQGAP1 interaction with active Rac1 (Rac1-GTP) is inhibited by Ca^2+^/calmodulin. The Ras GAP-related domain (GRD), RASGAP C-terminal (RGCT) and C-terminal (CT) regions of IQGAP1 bind to the switch I and switch II domains of Rac1-GTP (blue lines) to maintain Rac1 in its active state. Ca^2+^/calmodulin binds to the four isoleucine/glutamine-containing (IQ) motifs (thick red line) within IQGAP1, which abrogates IQGAP1/Rac1 interaction. In addition, Ca^2+^/calmodulin binds to the N-terminal calponin homology domain (CHD) (red line) of IQGAP1. This impairs IQGAP1 interaction with F-actin, blocking IQGAP1 from stimulating F-actin crosslinking, bundling, and capping. The CHD, coiled-coil repeat (CC), tryptophan-containing proline-rich motif (WW), IQ, GRD, RGCT and CT domains of IQGAP1 are indicated (see Section 4.2.2 for details) (**C**) Ca^2+^/calmodulin interacts and activates PI3K. The N- and C-terminal lobes and the flexible central linker of Ca^2+^/calmodulin bind to the N-terminal (nSH2) and C-terminal (cSH2) domains of the p85 subunit of PI3K (blue lines). This interaction releases the p85-mediated autoinhibition of the catalytic p110 subunit of PI3K. Activated PI3K then phosphorylates phosphatidylinositol-4,5-biphosphate (PI(4,5)P_2_) and generates phosphatidylinositol-3,4,5-triphosphate (PI(3,4,5)P_3_), which can bind and activate several Rac1 guanine nucleotide exchange factors (Rac1 GEFs) to increase Rac1-GTP levels (see Section 4.3 for further details).

**Figure 5 ijms-21-03680-f005:**
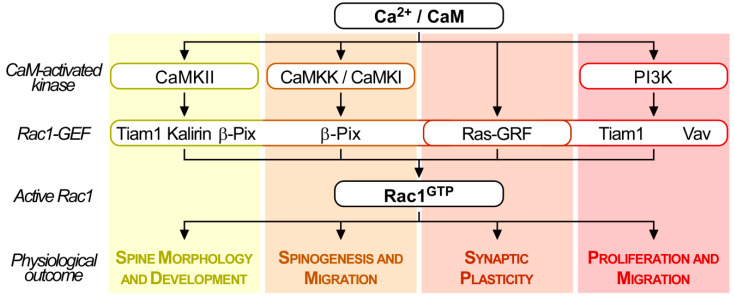
**Calmodulin-mediated activation of Rac1-specific GEFs controls multiple Rac1-dependent cellular functions.** This scheme highlights the ability of Ca^2+^/calmodulin (Ca^2+^/CaM) to bind and activate several kinases, including calcium/calmodulin-dependent protein kinase I and II (CaMKI, CaMKII), CaMK kinase (CaMKK) and phosphoinositide 3-kinase (PI3K). This leads to the activation of Rac1-specific guanine nucleotide exchange factors (Rac1-GEFs), including T-lymphoma invasion and metastasis-inducing protein 1 (Tiam1), PAK-interacting exchange factor β (β-PIX), kalirin, Ras guanine nucleotide releasing factor (RasGRF), and Vav. These Rac1-GEFs then promote GTP loading of Rac1, promoting Rac1 activation, which is associated with fundamental cellular activities, including spine morphology and development, spinogenesis, synaptic plasticity, proliferation, and migration (see text for further details).

**Table 1 ijms-21-03680-t001:** Small GTPases interacting with calmodulin. The various GTPases, their functions, the methodologies to assess their interaction with calmodulin, the cell lines analyzed, the calmodulin function as well as their interacting domains or protein sequences are listed. The relevant references are also given. Abbreviations: AC, affinity chromatography; BRET, bioluminescence resonance energy transfer; CBR, calmodulin binding region (basic amino acids are in red); CoIP, coimmunoprecipitation; CompMod, computational modelling; FS, fluorescence spectrometry; PXL, peptide cross-linking; RC, reconstituted complex; SF-TAP, Strep/Flag tandem affinity purification; YTH, yeast two hybrid.

GTPase	Function	Interaction method	Species	Calmodulin (CaM) action	Refs
**Cdc42**	Filopodia formation and cell-cycle progression	CoIP, AC, RC	Human (platelets)	Cdc42 inhibition CBR:^151^AVKYVECSALTQK^164^ (hypothetical amphipathic α-helix)	[23]
		AC	Human (MCF-7)	Regulation of Cdc42 signalling through IQGAP1	[48]
**Kir/Gem**	Cell proliferation, Neurite extension and flattening (Gem); invasion and metastasis (Kir). Inhibition of Ca^2+^ channels (Kir/Gem)	FS, RC	Synthetic peptides, in vitro	Kir/Gem inactivation (inhibits GTP binding), regulates intracellular localization CBR:^264/265^KARRFWGKIVAKNNKNMAFKLKSKS^288/289^ (hypothetical amphipathic α-helix)	[31]
			Xenopus oocytes	Inhibition of high voltage-activated calcium channels and regulation of intracellular distribution	[63]
			Monkey (COS-1)	Regulate intracellular localization between cytoplasm and nucleus, in a complex with 14-3-3	[61]
		CoIP, RC	Monkey (COS-1)	Inhibits association with importin α5 and nuclear translocation	[66]
**KRas-4B**	Cell proliferation, differentiation, survival and apoptosis	CoIP, AC, RC	Mouse (NIH3T3)	KRas inactivation	[19]
		RC, YTH.	Human (platelets)	Down-regulation of KRas signalling and KRas membrane transport	[20]
		AC	Mouse (NIH3T3)	CBR:^151^GVDDAFYTLVREIRKH^166^ (amphipathic α-helix5) CBR: C-terminal prenyl group	[16]
		RC		CBR:^170^KMSKDGKKKKKSKTKC^185^+ prenyl CBR:^180^KSKTKC^185^+prenyl group	[67,68]
		AC	Mouse (NIH3T3)	Inhibits KRas Ser181 phosphorylation	[29]
**Rab3A**	Synaptic and axonal transport. Regulated exocytosis in neuronal and endocrine cells	PXL	Rat brain synaptosomes	Dissociation of Rab3A from synaptic membranes CBR:^62^KTIYRNDKRIKLQIWDTA GQERYR^85^	[36,69]
			Rat (PC12)	Inhibition of exocytosis	[30]
		CoIP, PXL	Rat (pancreatic islets)	Inhibits insulin secretion	[42]
			Rat brain synaptosomes	Promotes GTP binding	[70]
			Human (spermatozoa from sperm donors)	Acrosomal exocytosis	[71]
**Rab3B**	Calcium-dependent exocytosis	AC	Mouse brain		[10]
		RC	Human (platelets)	Ca^2+^-dependent secretion in platelets	[33]
**Rab3D**	Maintenance of osteoclastic border membrane	AC	Mouse brain		[10]
		YTH, BRET, RC	Monkey (COS-1, reconstituted complex)	Promotes osteoclastic bone resorption	[46]
**Rabs: 6B, 8B,10, 15, 33B, 37.**		AC	Mouse brain		[10]
**Rac1**	Cytoskeletal organization, migration, adhesion, proliferation, endocytosis, vesicular trafficking	CoIP, AC, RC	Human (platelets)	Rac1 activation CBR:^151^AVKYLECSALTQRG^164^ (hypothetical amphipathic α-helix)	[23]
		AC, RC, CompMod	Human (HeLa)	Rac1 activation	[21]
		AC	Monkey (COS-1)	Rac1 activation Impairs binding to PIP5K CBR:^171^EAIRAVLCPPPVKKRKRK^188^ and the C-terminal prenyl group	[22]
		SF-TAP	Human (HEK293T)		[72]
**Rad**	Skeletal muscle motor function, cytoskeletal organization and glucose transport	RC	GST-Rad incubated with CaM-sepharose in vitro	Regulates intracellular localization CBR:^278^AKRFLGRIVARNSRKMA FRA^297^ (hypothetical amphipathic α-helix)	[31]
					[65]
**RalA**	Cell proliferation, migration, filopodia formation, differentiation, cytoskeletal organization, vesicular transport, exocytosis and receptor endocytosis. Synaptic and axonal transport	AC, RC, YTH	Human (platelets)	RalA activation	[38]
		RC	Human (HeLa)	RalA activation CBR: C-terminal prenyl group	[39]
			Rat brain synaptosomes	Dissociation of RalA from synaptic membranes	[69]
			Purified RalA, in vitro	Regulates GTP binding CBR:^183^SKEKNGKKKRKSLAKRIR^200^ (hypothetical amphipathic α-helix)	[34,40]
			Rat brain synaptosomes	Regulates GTP binding	[73]
**RalB**	Cell proliferation, oncogenic transformation	CoIP, AC, RC, YTH	Human (platelets)	RalB activation CBR: C-terminal prenyl group	[38,39]
			Rat brain synaptosomes	Regulates GTP binding	[73]
**Rem**	Ca^2+^channel regulation	CoIP, AC	Human (HEK293, TSA20)	No Ca^2+^ channel inhibition CBR:^165^QRARRFLARLTARSARRR^282^	[74]
**Ric**		Dot-blot, AC, CoIP	Drosophila	Regulation of Ca^2+^-mediated neuronal signal transduction CBR:^242^RRSRWWRIRSIFALVFRR RR^261^	[35]
			Drosophila (cross of Ric mutants with CaM mutants)	Ric inhibition	[56]
**Rin**	Ca^2+^ signalling in neurons	Dot-blot	Mouse	Regulation of calcium-mediated neuronal signal transduction CBR:^194^RKLKRKDSLWKKIKASLKKKRENML^218^	[32]
		CoIP	Monkey (COS-7)	Regulation of Rin activation	[54]
			Rat (PC12)	Induce neurite outgrowth	[55]

**Table 2 ijms-21-03680-t002:** **Effect of calmodulin on MAPK and Rac1 activation in different cell lines.** The impact of calmodulin on the activity of the MAPK pathway in different cell lines using calmodulin-specific inhibitors is shown. Some of the cell lines listed here, often in separate studies, have also been analyzed for Rac1 signalling and wherever possible, the outcome of Ca^2+^/calmodulin mediated activation of Rac1 is also listed. See text for further details. (-, inhibition; +, activation; ND, not determined).

Cell Type (Species)	Ca^2+^/Calmodulin Effects	
	MAPK Pathway (Refs)	Rac1 Activity (Refs)
NIH3T3 fibroblasts (mouse)	*-* [144]	*+* [157]
Swiss3T3 fibroblasts (mouse)	*-* [19]	*+* [158]
HeLa (cervix carcinoma, human)	*-* [145]	*+* [21]
NRK (kidney epithelial, rat)	*-* [144]	ND
A431 (epidermoid carcinoma, human)	*-* [142]	ND
COS1 fibroblasts (kidney, monkey)	*+* [143]	*+* [22]
CHO (epithelial ovary, hamster)	*+* [143]	ND
HEK293 (embryonic kidney, human)	*+* [142]	ND
PC12 (adrenal phaeochromocytoma, rat)	*+* [156]	ND
Primary hepatocytes (rat)	*+* [155]	ND

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
