# Peer review of "Pleiotropic Roles of Calmodulin in the Regulation of KRas and Rac1 GTPases: Functional Diversity in Health and Disease"

_ijms, 2020, doi:10.3390/ijms21103680_

Round 1
Reviewer 1 Report
This paper provides a compendium of findings describing the involvement of Calmodulin in the regulation of KRas and Rac1 GTPase. This review is both useful and informative. The authors adequately organized the review and provided general information on various pathways related to KRas and Rac1 to put in perspective Calmodulin global effect. The potential therapeutic impact of Calmodulin inhibitors as anti-cancer agents is also highlighted. This is a very nice effort to organize the complex and sometimes confusing literature around this topic. However, the authors should be careful not to over-extrapolate unproven hypotheses.
Q1: One of the conclusions which is carried throughout this review and incorporated in figure 1 and 2 is that Calmodulin interacts preferentially with (active) KRas. Recent in-vitro experiments clearly contradict this statement (reference 109). It should be stated in chapter 3.1 that though there might be differences between in-vitro and in-cell (lysates) experiments, evidence for a preferential interaction of Calmodulin with active KRas in cells are still lacking and further experiments are required to validate this statement.
Q2: The statement line 443 seems to imply that KRas phosphorylation at S181 is required for Raf1 activation. The sentence should be modified to avoid confusion.
Q3: What is the experimental evidence of the statement made line 443-444 and repeated 581-584? Why would GAPs be particularly active on Calmodulin/KRas complexes or specific microdomains? Calmodulin/KRas interaction does affect MAPK signaling but accordiong to the reviewer the cause is most likely not related to S181 phosphorylation. Many studies suggest that it reduces MAPK signaling mainly by disrupting KRas from the membrane thus preventing its signaling (references 20, 81, 109, etc.).
Q4: KRas phosphorylation at S181 is one of the main points discussed. The reviewer is not aware of precise detection of KRas 181 phosphorylation (i.e. not a comassie gel) ever being reported in the literature. Unless the authors can provide a clear experimental evidence of this event taking place in cells such as top-down proteomics, they should be careful to present these mechanisms of actions as hypotheses to be proven and not facts.
Q5: Line 541 suggests that S181 phosphorylation prevents calmodulin binding. Reference 81 shows that a phosphomimetic mutant S181E has no effect on preventing Calmodulin from interfering with KRas membrane binding.
Q6: Line 524-529 describe a mechanism by which KRas S181P would segregate in nanodomains enriched in PI3K and Raf1. These results, although compelling, are based on a series of indirect measurements that require further experimental validation.
Q7: The cell line dependent effect of calmodulin on Kras signaling raised between line 485-495 should be reflected in a table displaying calmodulin inhibitory or activating effect on MAPK. In the same table the calmodulin dual effect on Rac1 signaling should be displayed activation (i.e. summarized in figure3) inhibition (IQGAP1).
Q8: Line 504 states that Ras is activated by PKC. Did the authors mean phosphorylated?
Q9: Line 510-511 it is suggested that KRas S181P is mostly relocated from the plasma membrane to the endocytic compartment. Many studies, done with phosphomimetic mutant show that KRas S181E or D is mostly at the membrane (reference 121, 81). This sentence should be modified: “causing partial relocation…).
Q10: Minor errors in words of phrases could be corrected:
Line 854: effecto should be effector
Reviewer 2 Report
The authors have set out to provide important insight into a perplexing problem: how does calmodulin modulate so many events simultaneously, some of which are contradictory? It is important to understand this if calmodulin is to be an effective target in the fight against cancer and other diseases in which this small protein plays such a significant role. While the authors have surveyed the literature and put a collection of results together, the review doesn’t flow well and gets bogged down in the weak writing.
Line 54. The authors use the term “chapter” from here on when I think they mean “section”
Generally, reviews are supposed to compile the literature in order to communicate the state of the art to others. One key element in a review is clarity. This element is missing from this review.
The authors have a tendency to be confusing by trying to include too much information in a single sentence. For example, Lines 105-108. “Upon enhanced glucose levels and the subsequent rise in Ca2+ levels in pancreatic β-cells, the dissociation of Ca2+/calmodulin from Rab3A due to competition with other CaMBPs with a higher calmodulin affinity, for example Ca2+/calmodulin activated protein kinase II (CaMKII), and their involvement in granule secretion could be the underlying cause for increased insulin release [42].” The reader shouldn’t have to figure out what is being said.
The authors use words that undermine what they are trying to say. For example: Lines 113-115. “Rac1 and Cdc42 also represent GTPases of the Ras superfamily that are considered CaMBPs [22, 23]. Similar to the calmodulin/KRas interaction modules [16], several domains facilitate these protein communications.” On line 75 the authors stae Rac1 binds to calmodulin and on lines 83-84 both Rac1 and cdc42 are referred to as calcium-dependent CaMBPs. So why are they now “considered” to be CaMBPs and what does the term “communications” mean? Is this supposed to be another word for binding? The way these sentences are written it undermines the content of the review. If there are concerns about the binding of any of the GTPases to calmodulin then it should be clarified at the start. If there are no concerns then the authors need to be consistent in their statements and not undermine themselves with casual phrasing.
Why is Table 1 not referred to in the text?
Why not include CaM binding domains in Table 1 on the calmodulin binding attributes of the small GTPases? Are there any conserved CaM binding regions, domains between the various GTPases?
Section 3. Line 204-248. This is an exceedingly long paragraph that covers a diversity of concepts. One topic that is covered is Ras mutations but nothing is said about the relevance of these mutations to CaM-binding or to regulation by CaM. In fact, the content of section 3 only becomes relevant starting in 3.1 where the role of CaM binding to KRas is introduced, making the previous 3-4 paragraphs overdone since they are of little importance to the aim of this review.
The whole section on KRas/CaM binding (3.1 on) lacks any insight into the binding region—what kind of binding domain is it? What is its sequence? That’s important if it is to be used to make inhibitors or peptides directed against it.
Section 3.2 on. I think a figure showing the potential interaction between CaM, the CaM-binding domain in KRas, farnesyl group, nucleotide binding and its significance (e.g., localization of KRas, other?) would help the reader wade through this content. This is summarized somewhat as part of Figure 3 but the details are missing there.
Line 330-332. “Together with findings from several other studies, this suggested that calmodulin could participate as a molecular chaperone to transport KRas, independent of its GTP- or GDP-bound form, from the ER to the plasma membrane”. The authors improperly use the term “chaperone” which in molecular terms refers to proteins that mediate protein folding/unfolding, not the transport of molecule from one region to another. Possibly they meant “chauffeur”. This is another example of incorrect word usage that undermines what is potentially a valuable review.
Section 4. As for KRas, what is the binding domain sequence for Rac1? Do any mutations fall in the CaMBD? A detailed figure of the CaM/Rac1/GEF and other such as IQGAG-binding interaction and function (as suggested for KRas above) would assist the reader in understanding this complex interaction.
Lines 741-746. IQ motifs are typically understood to be Apo-CaM binding domains—this is not made clear by the authors who suggest that Ca-dependent CaM binding is expected.
Figure 3 (line 841). What is the significance of the four, color panels in Figure 1? Do their colors relate to the event associated with them? Should be explained in the figure legend.
Line 854. Heading “5. Adding Another Layer of Complexity: Rac1 is a KRas Effecto.” Insert “r” before period (i.e., Effector”.)
In summary, the authors have pulled together a large amount of data but fail to organize and discuss it in an easy to follow sequence. Maybe some subsections with subtitles, shorter accurate sentences and shorter more cohesive paragraphs would help.

Reviewer 3 Report
The review at hand deals with the roles of calmodulin in the regulation of small GTPases. It distinguishes from other reviews on similar subjects by focusing on the regulatory interactions of the calcium-binding protein with specifically two of GTPases, namely KRas and Rac1, basically in mammalian cells. As such, it gives an extensive overview of the latest literature and is generally well written. In my opinion, some of the descriptions of how different methods were employed to reach the conclusions summarized are often too lengthy and could be substantially shortened. However, this is a question of style and should be left to the authors discretion.
Also, the manuscript looks like it was only partially written by the native speaker. In several instances, “this” should read “these” and frequently past and present tense are mixed-up within one sentence. Please correct.
Apart from that, I have only a few minor editorial points:
- The constant references to previous sections (“abovementioned”) and various repetitions (e.g. on the explanations of the activation/deactivation cycles of GTPases by GEFs and GAPs, which is textbook knowledge) are very time-consuming for the reader and do not add any new information. I would strongly suggest to carefully go through the manuscript and eliminate most of these references.
- A lot of “fill-words”, like however, furthermore, on the other hand etc., are not necessary and frequently even misplaced, e.g. if there was not first hand remark. Please check and correct.
- Species names (Drosophila, Escherichia coli) should be written in italics.
- Formatting of Table 1 makes it a little confusing. I would suggest to eliminate the names of the authors in the references and just leave the numbers. Also, CoIPs, YTH, ACs etc. may be abbreviated accordingly, with explanations in the legend. This will provide more room for the column on Calmodulin action. This way, the Table should fit onto one or two pages instead of four, which makes it easier to capture.
- lines 302-308: First it is stated that KRas is not palmitoylated, then the authors talk about “depalmitoylated KRas-4A”. How is it possible to remove a modification from a protein, that does not exist in the first place?
- In both figures, protein names have the same colours like the background, i.e. green in green and pink in pink, simply with different shadings. This is very hard to differentiate. Please use black letters to designate the protein names. The same is true for the boxes describing the general outcomes.
- lines 455 to 468: Here are some examples, how English could be improved. “With regard to the impact of calmodulin”; Ca2+/calmodulin binding to their N-terminal IQ motif; ... thereby inhibits Ras and ...; ... calmodulin can activate both RasGRF1 and ...; ... Rac1 signalling, which thereby promotes long-term depression.
- line 549: suppressed
- Figure 2: I would suggest to draw KRas-GTP P-Ser181 away from the membrane, rather than anchored there. To my understanding the phosphorylation promotes dissociation from the membrane?
- line 581: There seems to be missing something in the sentence starting with “Complex formation of calmodulin ...”?
- line 680: “wild type” should be either written as such if it stands alone, or as “wild-type” if it refers to cells, genes etc.. There are all variations of this spelling in the manuscript. Please check carefully.
- lines 775-778: can be omitted completely.
- line 790: “contribute to impact on” should be replaced by e.g. “modulate”
- line 805: please correct English “implicating Tiam1 essential”?
- line 833: substitute for “deamination”
- line 847: Here, the authors talk of “guanine nucleotide exchange factors”, which I believe to be correct; in a previous part, they are only designated as “guanine exchange factors”
- line 854: ... Effector
Round 2
Reviewer 1 Report
Thank you for adressing all my concerns.
Reviewer 2 Report
The authors have done an admirable job of improving their manuscript. Not only did they address my concerns they have added valuable additional information (Table 1) and new insight with well done figures. This will be a valuable review for your readers.